ecology

Mediterranean sea, population collapse, temperate reefs, marine heatwaves, climate change, gorgonians

**Author for correspondence:**
D. Gómez-Gras
e-mail: danielgomez@icm.csic.es

# Population collapse of habitat-forming species in the Mediterranean: a long-term study of gorgonian populations affected by recurrent marine heatwaves

D. Gómez-Gras[1,2], C. Linares[2], A. López-Sanz[1], R. Amate[1], J. B. Ledoux[3], N. Bensoussan[1,4], P. Drap[5], O. Bianchimani[6], C. Marschal[7,8], O. Torrents[1], F. Zuberer[9], E. Cebrian[10,11], N. Teixidó[12,13], M. Zabala[2], S. Kipson[14], D. K. Kersting[2], I. Montero-Serra[2], M. Pagès-Escolà[2], A. Medrano[2], M. Frleta-Valić[1], D. Dimarchopoulou[15,16], P. López-Sendino[1] and J. Garrabou[1,4]

[1]Departament de Biologia Marina, Institut de Ciències del Mar (CSIC), Barcelona, Spain
[2]Departament de Biologia Evolutiva, Ecologia i Ciències Ambientals, Institut de Recerca de la Biodiversitat (IRBio), Universitat de Barcelona, Barcelona, Spain
[3]CIIMAR/CIMAR, Centro Interdisciplinar de Investigação Marinha e Ambiental, Universidade do Porto, Porto, Portugal
[4]Aix Marseille Université, CNRS, IRD, MIO, Université de Toulon, Marseille, France
[5]Aix-Marseille Université, CNRS, LIS-UMR, Université de Toulon, Marseille, France
[6]Septentrion Environnement, Marseille, France
[7]Institut Méditerranéen de Biodiversité et d'Écologie marine et continentale (IMBE), Marseille, France
[8]CNRS UMR, DIMAR, Centre d'Oceanologie de Marseille, Station Marine d'Endoume, Aix-Marseille Université, Marseille, France
[9]Centre de Recherches Insulaires et Observatoire de l'Environnement (CRIOBE), Moorea, Papeete, French Polynesia
[10]GR MAR, Institut d'Ecologia Aquàtica, Facultat de Ciències, Universitat de Girona, Girona, Spain
[11]CEAB-CSIC Centre d'Estudis Avançats de Blanes Blanes, Spain
[12]Sorbonne Université, CNRS, Laboratoire d'Océanographie de Villefranche, Villefranche-sur-Mer, France
[13]Stazione Zoologica Anton Dohrn, Ischia Marine Centre, Punta San Pietro, Ischia, Naples, Italy
[14]Department of Biology, Faculty of Sciences, University of Zagreb, Zagreb, Croatia
[15]Department of Zoology, School of Biology, Aristotle University of Thessaloniki, Thessaloniki, Greece
[16]Department of Fisheries, Animal and Veterinary Sciences, College of the Environmentand Life Sciences, University of Rhode Island, Kingston, RI, USA

DG-G, 0000-0002-9738-6522; CL, 0000-0003-3855-2743; AL-S, 0000-0002-4798-4524; JBL, 0000-0001-8796-6163; PD, 0000-0003-0528-9280; EC, 0000-0001-7588-0135; NT, 0000-0001-9286-2852; DKK, 0000-0002-2049-7849; IM-S, 0000-0003-0284-0591; MP-E, 0000-0003-1782-5124; AM, 0000-0003-0513-9215; DD, 0000-0003-3412-3503; PL-S, 0000-0001-9786-8006; JG, 0000-0001-9900-7277

Understanding the resilience of temperate reefs to climate change requires exploring the recovery capacity of their habitat-forming species from recurrent marine heatwaves (MHWs). Here, we show that, in a Mediterranean highly enforced marine protected area established more than 40 years ago, habitat-forming octocoral populations that were first affected by a severe MHW in 2003 have not recovered after 15 years. Contrarily, they have followed collapse trajectories that have brought them to the brink of local ecological extinction. Since 2003, impacted populations of the red gorgonian *Paramuricea clavata* (Risso, 1826) and the red coral *Corallium rubrum* (Linnaeus, 1758) have followed different trends in terms of size structure, but a similar progressive reduction in density and biomass. Concurrently, recurrent MHWs were observed in the area during the 2003–2018 study period, which may have hindered populations recovery. The studied octocorals play a unique habitat-forming role in the coralligenous assemblages (i.e. reefs endemic to the Mediterranean Sea home to approximately 10%

of its species). Therefore, our results underpin the great risk that recurrent MHWs pose for the long-term integrity and functioning of these emblematic temperate reefs.

## 1. Introduction

The disturbance regimes of marine ecosystems are shifting worldwide due to anthropogenic climate change. In particular, marine heatwaves (MHWs), which are discrete prolonged anomalously warm water events, are becoming more frequent and severe, causing deleterious impacts (i.e. mass mortalities of marine organisms) across the oceans [1–3]. Among the most affected organisms are habitat-forming species, including reef-building corals, sponges, gorgonians, seagrasses and kelps [2–4]. These organisms are major contributors to the ecosystem structure and functioning (i.e. through the provision of habitat, food, shelter or via facilitation processes). Therefore, the long-term resilience of impacted ecosystems (i.e. their ability to absorb disturbances while maintaining their identity, key functions and eco-evolutionary processes [5,6]) will primarily depend on both their resistance and ability to recover.

In the Mediterranean, MHWs are frequently impacting the coralligenous assemblages, which are temperate reefs harbouring approximately 10% of Mediterranean species [3,7]. Within these reefs, the habitat-forming octocorals (e.g. *Paramuricea clavata* and *Corallium rubrum*) have been among the most affected organisms, providing excellent biological models to explore the resistance and recovery capacity. However, while our understanding of their resistance to MHWs is improving (e.g. [8–12]), their recovery capacity is less understood. On the one hand, the resistance of Mediterranean octocorals to MHWs is highly influenced by their physiology, which is in turn determined by the thermal conditions (i.e. optima and limits) to which they are adapted [13]. Consequently, MHWs can lead them to suffer acute physiological stress (e.g. via causing metabolic dysfunctions or energy shortages related to heat-induced increased respiration rates), reductions in fitness (e.g. via reducing their resistance to pathogens) and even death [11,14,15]. On the other hand, the ability of octocorals to recover mostly depends on the arrival of new settlers and on the growth and propagation of surviving individuals [16,17]. *Paramuricea clavata* and *C. rubrum* are long-lived species and accordingly show slow life histories (i.e. exhibit late sexual maturity (greater than 10 years), slow growth rates (less than 3 cm per year), low recruitment rates or high post-settlement mortality) and restrained dispersal capacity [18–23]. Hence, their recovery capacity from mortality events is low; especially when mortality of adult reproductive colonies across populations is high and the stock-recruitment dynamics are further diminished [24]. In such situations, recovery primarily depends on the regrowth of the partially damaged colonies, given that their slow growth rates can be protracted. Moreover, when Mediterranean octocorals experience partial mortality, the denuded skeleton surfaces are typically quickly overgrown by epibionts (e.g. filamentous algae, bryozoans, polychaetes). This colonization prevents tissue regrowth in damaged parts. Furthermore, it results in an extra weight and greater resistance to the water flow that may gradually lead to partial colony breakage or even total detachment. Consequently, partially damaged octocorals may keep suffering significant losses of their biomass years after a mortality event [10], and exploring their recovery capacity requires long-term monitoring.

In the summer of 2003, a severe MHW impacted the NW Mediterranean Sea with temperatures up to 3°C above the average [25]. During this event, several populations of *P. clavata* and *C. rubrum* suffered mass mortality [26]. Here, we aim to provide new insights into their recovery capacity by assessing the long-term (15 years) trajectory of five impacted populations from Scandola marine protected area (MPA), Corsica, France. Specifically, we first explored temporal changes in density and size structure, extent of injuries, percentage of affected colonies and biomass. Together, these indicators provide information about the recovery trajectories of the populations in terms of structure, dynamics and conservation of key ecosystem functions such as the provision of three-dimensional habitats [10]. Then, we calculated the biomass-density values of the populations over the years and represented them in relation to their species-specific self-thinning lines. The self-thinning line represents the dynamic equilibrium at crowding density in which further population growth in biomass depends on density decay (typically following a slope of −3/2 in undisturbed populations) [27,28]. Hence, this approach allowed us to investigate the long-term trajectories of the populations with respect to the ones that that they should have followed if remaining undisturbed [29]. Finally, since recovery in long-lived species may depend on the sustained absence of disturbances, we also explored if recurrent thermal stress conditions have occurred in Scandola since 2003. Overall, our results provide insights into the trajectories that MHW-impacted populations of Mediterranean octocorals could follow in the face of climate change. Accordingly, this study takes us a step towards understanding the role of climate change as a driver of change in temperate reefs.

## 2. Material and methods

### (a) Study area and temperature data

The Scandola MPA was established in 1975 and acknowledged by UNESCO as a world heritage site in 1980. Located off the NW coast of Corsica, it is remote from industries or dense human populations. Most human activities are prohibited within its waters with the exception of scientific diving and boating. Thus, in the absence also of any other apparent disturbance during the study period, MHWs were detected as the most significant source of disturbance for the monitored populations.

Detailed information on the extreme warm conditions experienced in Scandola during the summer of 2003 and their relation to mass mortality can be found in the previous studies [25,26]. Here, we focused on exploring (i) if MHWs have increased in frequency over the last decades in Scandola and (ii) if monitored populations have been recurrently exposed to heat stress after being firstly impacted in 2003. To answer the first question, we analysed sea surface temperature (SST) trends over the satellite era (1982–2018) by extracting SST data from the nearest available pixel from Scandola in the CMEMS Mediterranean dataset [30]. From this dataset, the total number of MHW days per year during the warm hydrological period (June to the end of November; JJASON) were calculated with respect to the 30-year climatological period (1982–2011) following the MHW-detection method proposed by [1], which considers MHWs as discrete events with temperatures warmer than the 90th percentile (based on local climatology) for 5 days or more. For the second question, we used *in situ*, hourly, multi-year temperature data obtained for Scandola at 20 m depth based on a standard protocol implemented by the T-MEDNet initiative [31]. From this dataset, we calculated the number of extreme heat days per year (i.e. days exceeding the

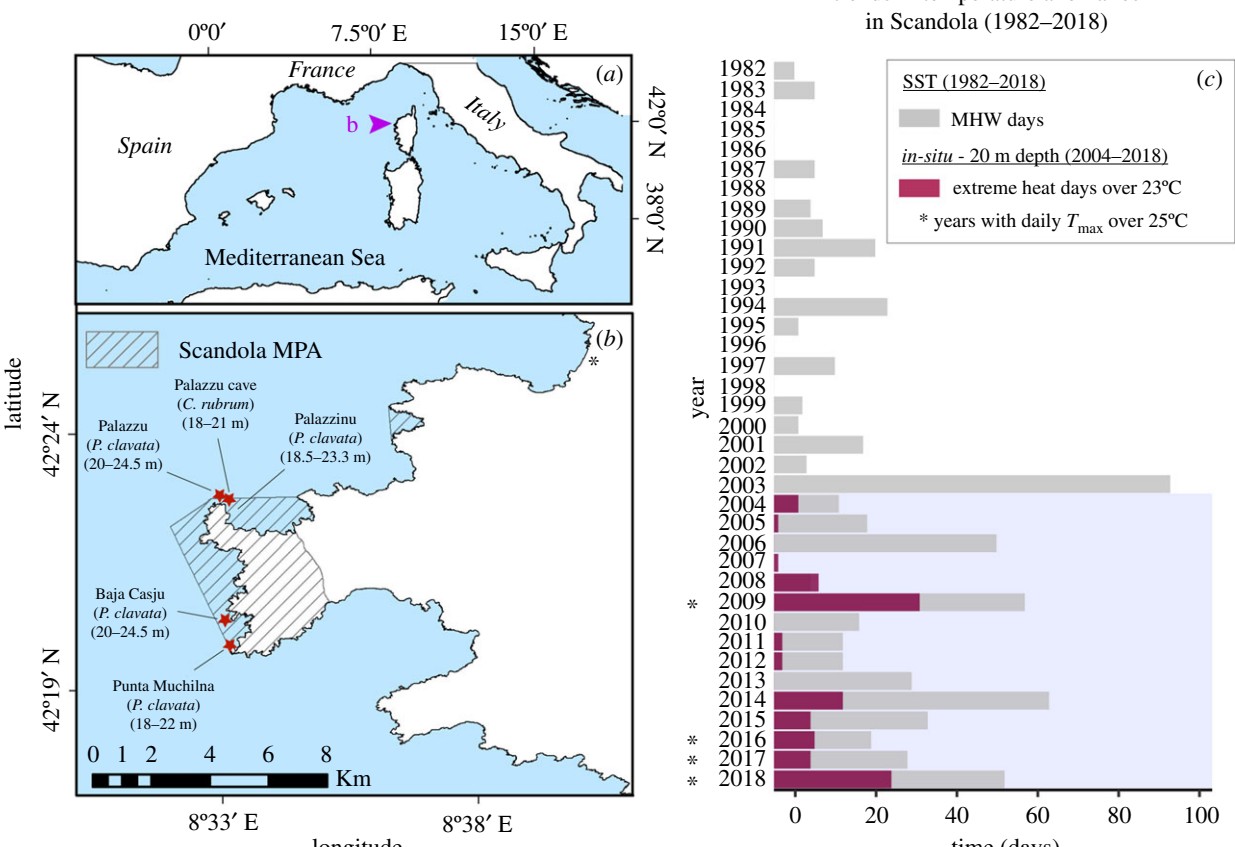

**Figure 1.** (*a*) Map showing the location of Scandola within the Mediterranean. (*b*) Location, species and depth of the five monitored populations within Scandola MPA. (*c*) Bars represent both the number of MHW days that occurred during the warm period (JJASON) of each year from 1982 until 2018 in the surface waters of Scandola and the number of *in situ* extreme heat days (those with T over the 90th percentile with respect to the local climatology) over 23℃ (sublethal threshold for the monitored species [11,12]) at 20 m depth. *In situ* T data were available from 2004 onwards (shadowed period). The years with *in situ* daily mean T reaching 25℃ (lethal threshold for the studied species [11,12]) are marked with asterisks (*). (Online version in colour.)

local inter-annual 90th percentile) and selected those reaching at least 23°C. We chose this latter indicator rather than *in situ* MHW days because at these temperatures, detrimental effects on the studied populations may occur before crossing the 5-consecutive-days threshold of the MHWs definition [11,12].

### (b) Population selection and quantitative monitoring

Different populations of *C. rubrum* and *P. clavata* have been monitored in Scandola MPA since 2003 for different scientific purposes. Here, we have selected only those that had exhibited clear signs of having been affected by the 2003 MHW, since we were interested in assessing their long-term recovery capacity. Overall, five impacted populations, ranging in depth from 18 to 24.5 m, met these criteria: four of *P. clavata*, and one of *C. rubrum* (figure 1*a,b*)

The five impacted populations were quantitatively surveyed for the first time after the 2003 MHW in May 2004. In the case of *P. clavata*, 15 to 30 quadrats (50 × 50 cm) were randomly placed by scuba divers at depths of 18 to 24.5 m depending on the population (see figure 1*b*). For *C. rubrum*, 24 permanent quadrats of 20 × 20 cm (0.96 m² in total) were set up at 18 to 21 m depth and photographically sampled. The sample size was selected in accordance to previous studies on these species [10,18,29,32]. Within each quadrat, we (i) counted the number of colonies, (ii) measured their maximum height (using a ruler for *P. clavata*, and the ARPENTEUR photogrammetric software [33] for *C. rubrum*) and (iii) estimated the extent and type of injury in every colony as the proportion of each colony's total surface with no tissue (i.e. denuded axis) or overgrowth by other organisms [29]. A total of 801 *P. clavata* colonies (more than 130 per

population) and 64 *C. rubrum* colonies were surveyed in May 2004. From these data, we calculated five population parameters: (i) the percentage (%) of affected colonies, measured as the proportion of colonies exhibiting more than 10% of injured surface with either recent or old injuries [10,29]; (ii) the mean percentage of extent of injured tissue, estimated as the average injured tissue per colony in each population; (iii) the density, calculated by averaging the number of living colonies found in each quadrat; (iv) the size structure, obtained by grouping the size of the measured colonies into different size classes (from 0 to 10, greater than 10 to 20, greater than 20 to 30, greater than 30 to 40, and greater than 40 cm height in *P. clavata* and from 0 to 3, greater than 3 to 6, greater than 6 to 9, greater than 9 to 15 and greater than 15 cm height in *C. rubrum*); and finally, (v) the live population biomass, calculated by averaging the cumulative live biomass found in each quadrat. Biomass was estimated previously for each *P. clavata* and *C. rubrum* colony according to the respective equations biomass (g) = 0.002 × height (cm)$^{2.61}$ [34], and biomass (g) = 0.1535 × height (cm)$^{1.9732}$ [32], and corrected for the proportion of injured surface as proposed by [10]. After the first quantitative surveys were conducted in May 2004, they were repeated in 2005, 2006, 2007, 2008 and 2017 for *P. clavata*, and annually until 2018 for *C. rubrum*. Colony height (and biomass) was re-assessed only in 2011, 2013, 2017 and 2018 for this latter species.

Finally, the reference values of the populations in 2003 (before the MHW) were estimated from the quantitative surveys of May 2004 by considering all tissue showing signs of old injuries (tissue overgrowth by organisms) as already injured before the event, and all tissue showing signs of recent injuries (e.g. denuded axis or recent epibiosis) as healthy before the event.

We could assume 2004's recent injuries as negligible in 2003 because: first, the annual field trips carried out by members of our team in Scandola since 1993 (including 2002) revealed no sign of recent or past mass mortality before 2003 (see electronic supplementary material, figure S1a). This is in agreement with the lack of impact reported by [9] during the mortality surveys conducted to assess the impact of the 1999 MHW in the area. Second, the first signs of mass mortality affecting octocorals in Scandola were described to have appeared in early autumn of 2003 following the 2003 MHW [26] (see electronic supplementary material, figure S1b). Last, these signs of mass mortality were still visible in May 2004 as recent injuries (i.e. denuded axis or recent epibiosis), further evidencing that the vast amount of recently injured tissue observed in May 2004 was unequivocally the consequence of the 2003 MHW.

## (c) Statistical analyses

We used generalized linear mixed models (GLMMs) to test the impact and recovery of the monitored octocorals from the 2003 MHW. Specifically, time (in years) was set as a fixed factor and the reference values (the ones from before the 2003 MHW) of affected colonies, density, biomass and size structure (dependent variables) were compared with the values obtained onwards (from 2004 until 2017–2018). Significant differences between 2003 and 2004 were considered as indicators of 2003 MHW impact, whereas a lack of difference thereafter (from 2005 to 2018) with respect to the 2003 values was considered as an indicator of recovery. After these models, we fitted a new set of GLMMs with the proportion of affected/dead colonies set as dependent variables and size class × time (2003 versus 2004) set as interacting fixed effects to test if the proportion of affected/dead colonies varied with size class due to the 2003 MHW. The family functions of the models were selected based on the nature of the dependent variables and after visually checking the most likely distribution of the data [35]. The models were fitted using the functions 'glmer' and 'clmm' from the lme4 and ordinal packages [36,37], and their overall performance was visually inspected by observing the residual distributions and quantile–quantile plots. Finally, post hoc Fisher's exact tests were used to explore if the size structure of each *P. clavata* population changed over time [38]. All analyses were performed using the software R v. 3.5.0 (R Core Developer Team, 2018).

## (d) Visualization of disturbance versus recovery in species with self-thinning growth

To further explore the trajectories of the monitored populations over the monitored periods with respect to the ones that they should have followed if remaining undisturbed or having totally recovered, we represented their log–log biomass against density values in relation to their species-specific self-thinning lines [29]. The self-thinning line for *P. clavata* had been calculated in a previous study [29] (slope of −1.45; $R^2 = 0.76$; $p < 0.01$), whereas the *C. rubrum* self-thinning line was calculated *de novo* in this study by fitting a linear regression to the biomass and density values of 28 undisturbed populations maturing at crowding density provided by [32] (slope of −1.44; $R^2 = 0.62$; $p < 0.01$; electronic supplementary material, table S1).

# 3. Results

## (a) Temperature trends in Scandola

The SST trends over the 1982–2018 period revealed a gradual increase in the average number of MHW days per year (figure 1c). Before 2003 (1982–2002 period), the average number of MHWs per year was $8.2 \pm 8.4$ (mean ± standard

deviation (s.d.)), whereas from 2003 (2004–2018 period), this value increased to $31.3 \pm 20.7$. Further, the *in situ* T trends revealed that populations were recurrently exposed to thermal stress conditions since 2003. In fact, three years (2009, 2014 and 2018) exhibited more than 10 extreme heat days over 23°C, while 4 years (2009, 2016, 2017 and 2018) exhibited at least one extreme heat day exceeding 25°C.

## (b) Percentage (%) of affected colonies and extent of injury

In *P. clavata* populations, the percentage of affected colonies significantly increased from $7.9 \pm 4.2\%$ (mean ± s.d.) before the 2003 MHW to $68.5 \pm 21.3\%$ in 2004 (figure 2a–d; electronic supplementary material, tables S2a and S4a). In particular, the percentage of recently affected colonies (those with more than 10% of denuded axis) ranged from 35.7% in Baja Casju to 86.1% in Punta Muchillina after the event. Furthermore, the mean percentage of extent of injury of colonies also peaked immediately after the 2003 MHW in the four populations (figure 2f–i; electronic supplementary material, table S2a). Overall, $40 \pm 17\%$ of colonies died during the event. During the following years, both the percentage of affected colonies and the extent of injury were gradually reduced in each population until 2008. Yet, the two indicators had increased again in 2017 in two out of the four populations (Baja Casju and Punta Muchillina).

In *C. rubrum*, the percentage of affected colonies exhibited a significant increase from 4.7 to 42.2% due to the 2003 MHW (figure 2e; electronic supplementary material, tables S2b and S5b), which coincided with a peak in the average extent of injury of colonies of 19.9% (figure 2j). During the following years, both the percentage of affected colonies and the extent of injury were gradually reduced until 2009, after which some newly affected colonies were detected. However, the highest increase in the percentage of affected colonies and extent of injury occurred in 2016, with a 41.7% and a 35.5% increase, respectively. In 2018, the cumulative percentage of affected colonies was 81.8%, while the extent of injury reached a value of 54.6%.

## (c) Density and biomass

The density of the *P. clavata* populations ranged between 23 and 44 gorgonians per m$^2$ (mean $32 \pm 9$ s.d.) before the 2003 MHW, while the biomass ranged between 106.7 and 347.7 g m$^{-2}$ (mean $240.4 \pm 100.2$ s.d., figure 2k–n, p–s; electronic supplementary material, table S3a). After the MHW, the density decreased in all populations until a mean value of $19 \pm 7$ s.d. colonies per m$^2$, resulting in a significant average density decay of a 39.7% (a minimum decrease of 20.1% in Baja Casju and a maximum of 54.9% in Palazzinu) in 2004 (figure 2k–n; electronic supplementary material, table S4b). Similarly, the biomass decreased in all populations until reaching a mean value of $80.7 \pm 33.8$ s.d. g m$^{-2}$, resulting in a significant average decrease of 62.7% (figure 2p–s; electronic supplementary material, table S4c). During the following years, neither the density nor the biomass recovered the initial values in any of the studied populations (figure 2k–n, p–s; electronic supplementary material, table S4b,c). At the end of the monitoring in 2017, populations densities ranged between 6 and 14 gorgonians per m$^2$ (mean $9 \pm 3$ s.d.), which represents an average relative loss of $70.7 \pm 7.7\%$ s.d. when compared to

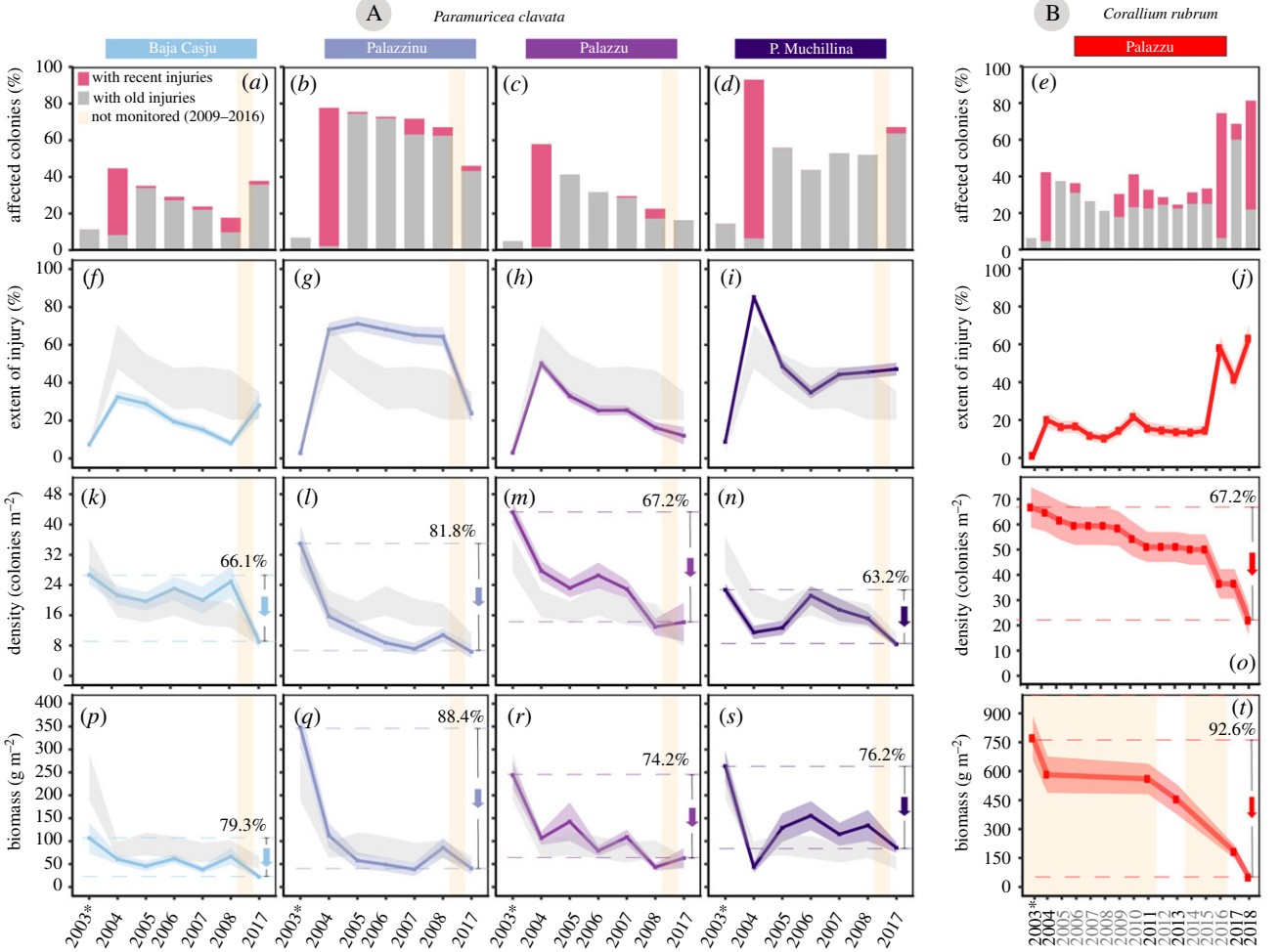

**Figure 2.** Temporal trends in the percentage (%) of affected colonies (*a–e*), percentage of extent of injury ± s.e. (*f–j*), density (number of colonies per m² ± s.e.) (*k–o*) and biomass (g per m² ± s.e.) (*p–t*) for the monitored octocoral populations. The cumulated relative density (or biomass) loss is shown at the right side of each plot. The light grey background polygons shown in (*f–i, k–n, p–s*) represent the average values ± s.e. of the four populations of *P. clavata*. (Online version in colour.)

the values of 2003. Similarly, biomass values had cumulative fallen on average by 79.6 ± 8.9% s.d.

For the *C. rubrum* population, the density slightly decreased due to the 2003 MHW from 67 ± 5 colonies per m² (mean ± s.e.) before the 2003 MHW to 65 ± 5 in 2004 (figure 2*o*; electronic supplementary material, table S3b). Since 2004, it continued to gradually decrease at an average rate of 1 colony per m² per year until 2014, when 50 ± 4 colonies per m² (mean ± s.e.) were found. In a similar tendency, the population biomass was also gradually reduced since the 2003 MHW, from 770.8 ± 113.3 g m⁻² (mean ± s.e.) to 450 ± 77.3 in 2013 (figure 2*t*; electronic supplementary material, table S3b). From 2016, both the density and biomass decreased sharply, until reaching minimum values of 22 ± 3 colonies per m² (mean ± s.e.) and 57.2 ± 18.75 g m⁻² (mean ± s.e.) in 2018. At this time, density and biomass had decreased on average by 67.2% and 96.2% respectively since 2003.

## (d) Size structure

All *P. clavata* populations presented a normal distribution of size frequency (Shapiro–Wilk; $p > 0.05$) in 2003, with a peak in the 10–20 cm size class (figure 3*a–d*). After the 2003 MHW, all populations maintained the same size structure (electronic supplementary material, figure S2a–d; Fisher's

test, $p > 0.05$). Accordingly, there were not significant differences in the percentage of dead colonies among size classes during the 2003 MHW ($\chi^2 = 11.2$, d.f. = 4, $p = 0.85$; electronic supplementary material, figure S3c,d). However, larger colonies were more affected than smaller ones ($\chi^2 = 49.5$, d.f. = 4, $p < 0.001$; electronic supplementary material, figure S3a,b). During the following years, the size frequency distributions were relatively stable (electronic supplementary material, figure S4a–d), and in 2017, the size frequency distribution was not significantly different from the one in 2003 for three of the four populations (figure 3*a–d*).

The *C. rubrum* population presented a non-normal distribution (Shapiro–Wilk; $p < 0.05$) in 2003, in which the different size classes until 15 cm were similarly represented (figure 3e). After the 2003 MHW, the population maintained roughly the same size structure (electronic supplementary material, figure S2e; Fisher's test, $p > 0.05$). However, smaller colonies were less affected than larger ones (although not significantly; electronic supplementary material, figure S5), and a progressive reduction in the frequency of colonies larger than 6 cm was observed since 2011 (electronic supplementary material, figure S4e). When comparing the size structure of the population in 2003 and 2018, significant differences among them and a new large peak in the 3 to 6 cm size class were found (figure 3e; electronic supplementary material, table S5d).

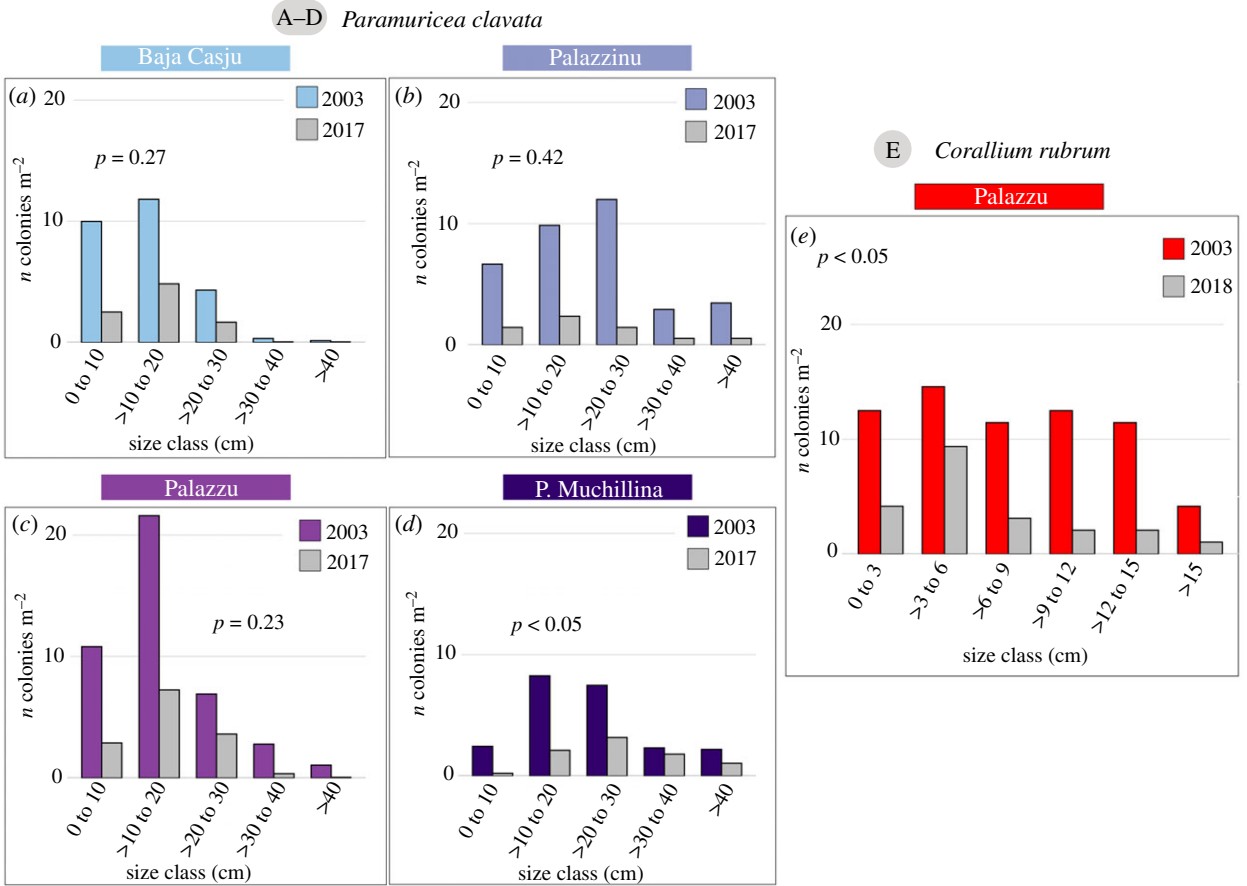

**Figure 3.** Size structure of the *P. clavata* (*a–d*) and *C. rubrum* (*e*) populations before the 2003 MHW and at the end of the monitoring period (2017–2018, respectively). (Online version in colour.)

## (e) Populations trajectories in relation to the species-specific self-thinning lines

Before the 2003 MHW, all monitored octocoral populations were closely related to their species-specific self-thinning lines, indicating that they were undisturbed (figure 4; electronic supplementary material, figures S6a–d and S7). After the 2003 MHW, all populations were shifted away from the line. During the following years, different episodes of density and biomass loss, regrowth and recruitment could be tracked (electronic supplementary material, figures S6a–d and S7). However, populations never returned to their initial position and tended to gradually drift further away from the line. Overall, all *P. clavata* populations followed the same global trajectory over the 2003–2017 period, which was mostly determined by a substantial decline in density and a slight loss of mean colony biomass. The red coral population followed a similar disturbance trend, but with a higher loss in the mean colony biomass with respect to the density loss.

## 4. Discussion

Predictions of shifts in the structure and functioning of temperate reefs under climate change are challenged by a poor understanding of the long-term recovery capacity from MHWs of their habitat-forming species. Here, we show that populations of two key Mediterranean habitat-forming octocorals that were originally impacted by the 2003 MHW in an old and well-enforced MPA have not recovered after 15 years. Contrarily, they have followed collapse trends that have brought them to the brink of ecological extinction.

## (a) Immediate and mid-term (2–5 years) effects of the 2003 marine heatwave

Our results indicate that MHWs occurred in Scandola already in the eighties. However, thermal conditions were likely not stressful enough to overcome the thermotolerance limits of the studied octocorals until 2003, since no signs of warming-induced mass mortality had been observed in the area before that year [26]. Moreover, before the 2003 MHW, all monitored populations exhibited values of density, biomass and injury in the range of those typically reported for healthy populations across the NW Mediterranean (e.g. [29,32,39]). Furthermore, they were maturing following a typical self-thinning growth trajectory expected for undisturbed populations. However, the 2003 extreme warm conditions (described in detail in the previous studies [25,26]) severely increased the number of partially and totally dead colonies for both species, driving the populations far below their expected self-thinning trajectories. Despite differences in the degree of impact among populations (e.g. percentage of affected colonies varied between 35.7 and 86.1%), the percentage of affected colonies and the extent of injury exhibited on average an eightfold and 14-fold increase in *P. clavata* populations respectively, and a ninefold and 19-fold increase in *C. rubrum*. Moreover, the density and biomass

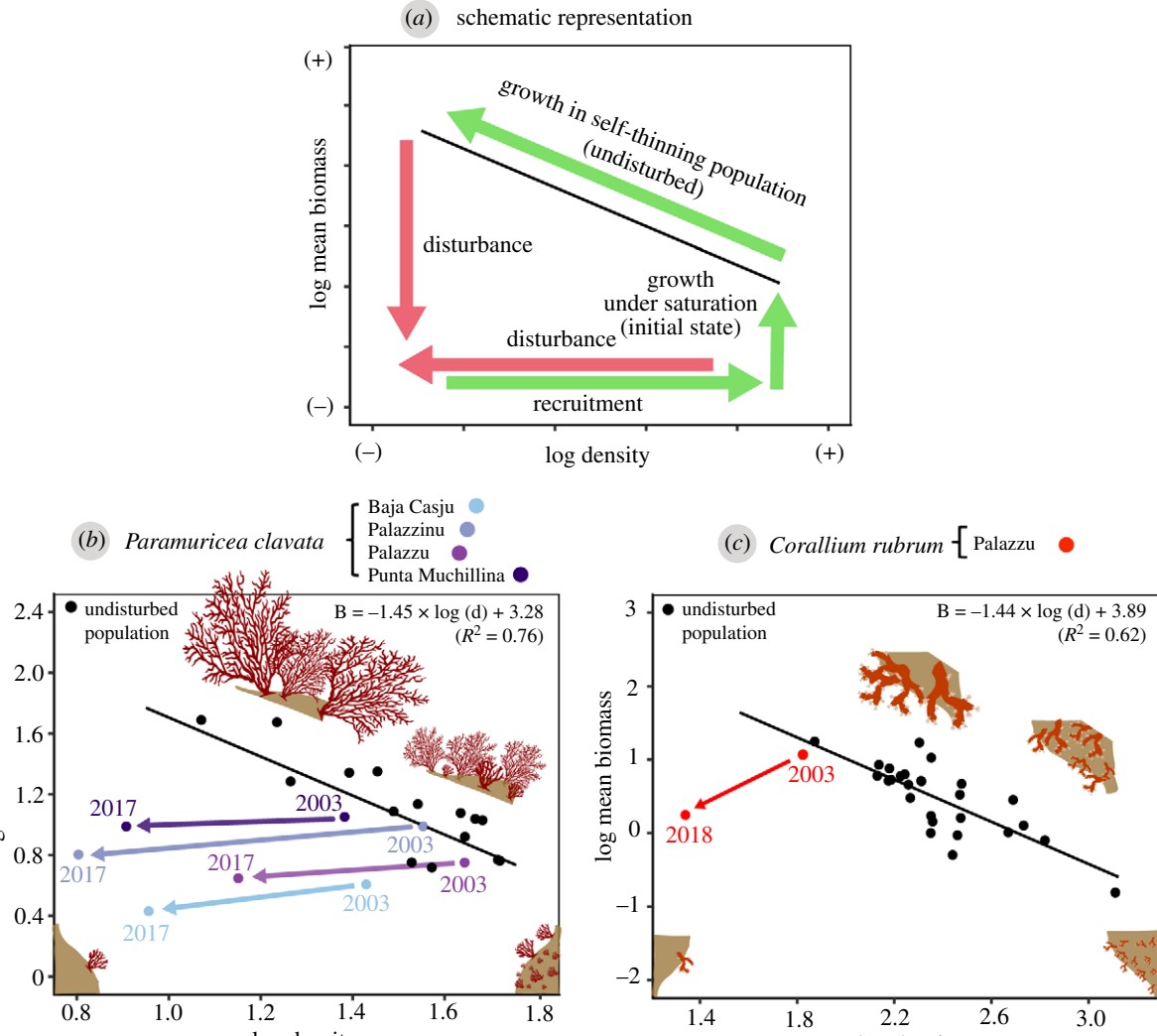

**Figure 4.** (*a*) Schematic representation of self-thinning growth in habitat-forming octocorals and the expected effects of disturbance and recovery (modified from [29]). From the lower right corner (foundation status), the growth of recruits moves the population vertically until it reaches the biomass saturation point. Above this point, the growth of colonies depends on a density decay due to intra-specific competition, resulting in populations which follow the self-thinning theoretical line (slope of −3/2), until reaching maturity [27,28]. Disturbed populations may be displaced from the self-thinning line due to loss in density or biomass. Recovery via recruitment or re-growth may result in a returning trajectory until potentially reaching the initial position; (*b,c*) show, respectively, the overall trajectory of the *P. clavata* and *C. rubrum* populations with respect to their species-specific calculated self-thinning lines. Only initial (pre-2003 MHW) and final (2017–2018 depending on the species) values are shown. For an extended version of this figure showing the complete trajectories, see electronic supplementary material, figures S6 and S7. (Online version in colour.)

decreased on average by about 40 and 64%, respectively, for *P. clavata*, and by about 3 and 25% for *C. rubrum*. These values are consistent with previous observations from MHW-induced mortality outbreaks of these and other octocorals across the Mediterranean (e.g. [3,8–10,26]), as well as from tropical hexacorals [4]. Interestingly, despite reductions in density and biomass, no significant changes in the size structure were observed in any population immediately after the 2003 MHW. For *P. clavata*, in which about 40% of the assessed colonies had died, this stability was driven by an overall size-independent mortality (electronic supplementary material, figure S3d). For *C. rubrum*, the size structure stability was the consequence of the low number of dead colonies (only two out of 64).

Overall, the 2003 MHW impacted Scandola during few months. Yet, its effects on octocoral populations extended for years. In 2008, *P. clavata* and *C. rubrum* populations had further reduced their density on average by 16% and 9%, respectively, from the values found immediately after the

2003 MHW. As no relevant mortality outbreaks were evident in these populations during these years (the percentage of recently affected colonies remained always low), the progressive decay in density during the 2003–2008 period was probably the consequence of the delayed mortality of the 2003's injured colonies, which was not compensated by recruitment. In coral species, sexual recruitment is essential to replenish disturbed populations [40]. Yet, sexual recruitment successful enough to attain recovery in the mid-term (2 to 5 years after the MHW) for *P. clavata* and *C. rubrum* is constrained. First, their recruitment rates are usually low, with pulses at decadal scales and high post-settlement mortality [18,20]. Second, they have restrained dispersal capacity, which hinders the arrival of settles from potentially non-impacted neighbour populations [22,23]. Lastly, the reproductive capacity of colonies may be further reduced in injured colonies, which may impair their stock-recruitment dynamics [24,41]. Accordingly, although we observed some new recruits in *P. clavata* populations in the 2003–2008

period, the overall number ($n = 137$) was not sufficient to counterbalance the number of dead colonies ($n = 962$). The same pattern was observed for *C. rubrum*, which showed an absence of recruitment despite 13% of the colonies having died in this period. Similar results were also obtained by [10] for *P. clavata*, reporting a number of recruits that barely reached 1/3 of the dead colonies over the first 4 years following a mortality outbreak. However, other studies found total density recovery in impacted populations of this species after just 3–7 years [16,42]. Nonetheless, to achieve a complete population recovery (i.e. bounce back to a state comparable to the one before the outbreak; *sensu* [43]), a complete recovery of biomass via successful growth or regeneration of survivors would be also essential. Given the slow growth rates of *P. clavata* and *C. rubrum*, achieving such a recovery in the mid-term seems very unlikely; even more so when injured colonies are usually subjected to high levels of epibiosis, which may considerably limit the likelihood of successful regeneration [10].

## (b) Long-term consequences (15 years) of the 2003 marine heatwave

All populations were farther from recovery in 2017–2018 than in 2008. On average and with respect to the 2003 values, the density and biomass had cumulatively decreased by around 71% and 80%, respectively, in *P. clavata* and by about 67% and 93% in *C. rubrum*. Thus, there was not only insufficient recruitment to compensate for the gradual loss of colonies, but the growth/regeneration of survivors was far from sufficient to offset the biomass loss. The size structure of *P. clavata* populations was mostly maintained, indicating that at least some large colonies resisted or recovered. By contrast, the *C. rubrum* population experienced a gradual shift towards the prevalence of small colonies (3–6 cm) due to the loss of necrosed parts in affected colonies over the years. However, regardless of the size structure trajectories, the general decline in large colonies (including reproductive ones) experienced by both species since 2003 has severely depleted their resilience and role as habitat-formers.

Overall, little doubt remains that the collapsing trajectories reported here had their origin in the strong 2003 MHW. Yet, the underlying factors and processes explaining the posterior collapse are still a matter of debate. A variety of intrinsic (genetic, physiological) and extrinsic (e.g. food availability, pathogens) factors have been evocated in the past as potential modulators of the short-term and long-term ecological consequences of MHWs (e.g. [8,15,25,26,41]). Yet, the fact that the populations were recurrently exposed to heat stress since 2003 in the apparent absence of other disturbances suggests that the temperature anomalies likely played the most decisive role in preventing the recovery. In line with this hypothesis, three recurrent mass mortalities (more than 10% of recently affected colonies) affecting the *C. rubrum* population were observed since 2003, and all occurred years with the highest heat conditions observed (2009, 2016 or 2018).

Nowadays, a growing body of evidence suggests that habitat-forming organisms are declining throughout the oceans in response to climate change, with countless tropical reefs, kelp forests and seagrass meadows suffering unprecedented losses following MHWs (e.g. [2–4]). Here, we add to this increasing evidence by showing that Mediterranean temperate octocorals could be also heading towards unprecedented declines due to a generally low recovery capacity from recurrent MHW impacts. Moreover, since our study was conducted in one of the oldest (greater than 40 years) and best enforced no-take MPAs in the NW Mediterranean, our results fuel the ongoing debate about the role of current MPAs in supporting the resilience of habitat-forming species in front of climate change [44]. Further research comparing long-term data from protected and non-protected areas should clarify if Mediterranean MPAs are reducing climate change impacts on these species to some extent. Yet, in our case, current protection was not enough. Without immediate action, further gorgonian losses throughout the Mediterranean could be expected, leading to less complex ecosystems and a potential great loss of associated functions and services [45]. The operationalization of a climate-responsive design and management of a fully protected network of MPAs could favour the preservation of benthic communities in some specific areas of the Mediterranean (i.e. potential climate refugia [46]). Yet, tackling climate change at its source by reducing $CO_2$ emissions remains pivotal for preventing irreversible change in the integrity and functioning of coral-dominated communities in the Mediterranean.

Data accessibility. All data and code supporting the results are available from the Dryad Digital Repository: https://doi.org/10.5061/dryad.18931zczk [47].

Authors' contributions. D.G.-G.: conceptualization, formal analysis, investigation, methodology, visualization and writing the original draft; C.L.: conceptualization, funding acquisition, investigation, project administration, supervision, writing the review and editing; A.L.-S.: methodology, writing the review and editing; R.A.: methodology, writing the review and editing; J.B.L.: investigation, writing the review and editing; N.B.: methodology, resources, writing the review and editing; P.D.: software, writing the review and editing; O.B.: investigation, writing the review and editing; C.M.: investigation, writing the review and editing; O.T.: investigation, writing the review and editing; F.Z.: investigation, writing the review and editing; E.C.: investigation, writing the review and editing; N.T.: investigation, writing the review and editing; M.Z.: investigation, writing the review and editing; S.K.: investigation, writing the review and editing; D.K.K.: investigation, writing the review and editing; I.M.-S.: investigation, writing the review and editing; M.P.-E.: investigation, writing the review and editing; A.M.: investigation, writing the review and editing; M.F.-V.: investigation, writing the review and editing; D.D.: investigation, writing the review and editing; P.L.-S.: investigation, writing the review and editing; J.G.: conceptualization, funding acquisition, investigation, project administration, writing the review and editing.

Competing interests. We declare we have no competing interests.

Funding. We acknowledge the 'Severo Ochoa Centre of Excellence' (CEX2019–000928-S) funding, the MCIU/AEI/FEDER [HEATMED; RTI2018-095346-B-I00], Interreg-Med Programme (5216 | 5MED18_3.2_M23_007 and 1MED15_3.2_M2_337), Foundation Prince Albert II Monaco [MIMOSA], the TOTAL-Foundation [Perfect] and the European Union's Horizon 2020 research and innovation programme (grant nos. 689518 and SEP-210597628). D.G.-G. is supported by a FPU15/05457 grant. C.L. acknowledges the support of ICREA. J.B.L is supported by the strategic Funding [UIDB/04423/2020 and UIDP/04423/2020]. N.T. received funding by the French National Research Agency-Make Our Planet Great Again [4Oceans-MOPGA and ANR-17-MPGA-0001]. D.K.K. is supported by a IJCI-2017-31457. D.G.-G., C.L., J.B.L., E.C., P.L.-S., D.K.K. and J.G. are part of the Medrecover group [2017 SGR 1521].

Acknowledgements. We thank the staff of the Scandola NR, especially chief-officer Jean-Marie Dominici, for facilitating optimum conditions to conduct the monitoring.

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
