## [Peer Review File · Proceedings of the Royal Society B: Biological Sciences]

Review History

RSPB-2020-2460.R0 (Original submission)

Review form: Reviewer 1 (Terry Hughes)

Recommendation

Accept with minor revision (please list in comments)

Scientific importance: Is the manuscript an original and important contribution to its field?

Good

General interest: Is the paper of sufficient general interest?

Good

Quality of the paper: Is the overall quality of the paper suitable?

Acceptable

Is the length of the paper justified?

Yes

Should the paper be seen by a specialist statistical reviewer?

No

Do you have any concerns about statistical analyses in this paper? If so, please specify them explicitly in your report.

No

It is a condition of publication that authors make their supporting data, code and materials available - either as supplementary material or hosted in an external repository. Please rate, if applicable, the supporting data on the following criteria.

Is it accessible?

Yes

Is it clear?

N/A

Is it adequate?

N/A

Do you have any ethical concerns with this paper?

No

Comments to the Author

The paper summarizes an interesting long-term study of the responses of marine benthic populations to recurrent heatwaves due to global warming. Although the study is spatially-limited (one marine protected area), it provides a rare insight into the long-term dynamics of structurally-important Mediterranean species. My comments are quite minor, to help improve the paper.

The paper is over-referenced, including quite a lot of self-citations. I would like to have seen a paragraph that compares these results to other studies in the Mediterranean, on coral reefs, kelp beds, etc. How general are these findings – are the same trends likely to be unfolding elsewhere?

Line 122. What does mucilaginous outbreaks mean? Disease?

Line 131. Hydrological period is unclear.

The site descriptions on Lines 136-139 are unclear. The site names appear in Fig. 2, but not in the methods. How far apart are they? Is Paluzzu cave the same as Paluzza? Were each of the four sites for *P. clavata* sampled at the same 18-26m depth range, or were some sites shallower than others? It appears to be the latter, from L39 (2.5m range). The authors sometimes seem to assume that readers will flip back and forth to the supplementary materials – but it would be better if the main text is self-explanatory.

L190. Saturated populations is unclear.

L272. Indistinctly?

The figures need improvement. Figure 1 should be taller. In Figure 2-4, the place names and cartoons on top are unnecessary. The font sizes are too small. Figure 3 should be presented as number of colonies per size class, not the proportion. The stability in proportions is misleading (L387-389), and understates the changes that have occurred. In particular, the absolute number of large, reproductive colonies must have declined substantially, compromising population

resilience.

I didn't find Figure 5 very useful, and the description of it on L313-314 is inadequate. I suggest deleting this figure.

Minor points:

Some of the English could be improved. For example:

L41. ...resilience to heatwaves

L89. Parsimonious is the wrong word.

L93.which can be protracted.

L104.by monitoring 5....

L108.resilience of coralligenous....

L114.both species also exhibit late.....

L220. % (here and elsewhere) should be percentage, or proportion.

L234. which coincided with a peak...

L250.neither the density nor...

L301, 303. Global?

L302.which was mostly determined by a substantial decline in density....

L347.is constrained.

L358.colonies having died....

L370. The last 4 words in the heading are not necessary.

L377.repeatedly impacted....

L387.was far from sufficient.....

Review form: Reviewer 2

Recommendation

Major revision is needed (please make suggestions in comments)

Scientific importance: Is the manuscript an original and important contribution to its field?

Good

General interest: Is the paper of sufficient general interest?

Acceptable

Quality of the paper: Is the overall quality of the paper suitable?

Good

Is the length of the paper justified?

Yes

Should the paper be seen by a specialist statistical reviewer?

No

Do you have any concerns about statistical analyses in this paper? If so, please specify them explicitly in your report.

Yes

It is a condition of publication that authors make their supporting data, code and materials available - either as supplementary material or hosted in an external repository. Please rate, if applicable, the supporting data on the following criteria.

Is it accessible?

Yes

Is it clear?

N/A

Is it adequate?

N/A

Do you have any ethical concerns with this paper?

No

Comments to the Author

Abstract

- Please define “coralligenous assemblages” for readers that have not encountered this term. Is this habitat limited to the Mediterranean?

Intro

- No mention of the mechanisms by which marine heat waves cause mortality of octocorals (and other habitat-forming species). Please dedicate a paragraph to outlining the physiological mechanisms of mortality.
- Line 88: “Habitat-forming species are long-lived”. But octocorals are colonial. Individual polyps are not long-lived, but the entire colony may be. Similarly, line 89 says recovery from mortality may be slow. But what about partial mortality (i.e., when part of colony survives?). In scleractinian corals, recovery can be very fast when mortality is only partial.
- Line 92: Why does recruitment not also contribute to population recovery? Unclear – please elaborate.
- Paragraph starting at line 112: Ok, this explains that recruitment potential is low for these species – please move this information to the Intro to provide context for the comment above.
- Paragraph starting at line 120: Is this region exposed to any sources of land-based runoff or other types of pollution? Seems unlikely that any coastal area in the Mediterranean has effectively banned “any local human stressors”.
- Line 122: “mucilaginous outbreaks”? please explain.
- Please provide a couple of sentences at the end of the intro summarizing what you measured and why.

Material and Methods

- Why were the various sites chosen? Are they characterized by same environmental conditions, human population densities, etc.? What is the explanation for the unbalanced sampling design (i.e., four sites for *Paramuricea clavata* but only one site for *Corallium rubrum*?). Why were multiple sites chosen?
 - Why were all these various aspects of the colonies measured (e.g., what are ecological or conservation implications of changes in density v biomass?) Please provide justification for study design before all of these different metrics are mentioned.
 - Line 157: “Finally, the status of the populations in summer 2003 (before the MHW) were derived from the data obtained in May 2004, considering all tissue showing signs of recent injuries (e.g., denuded axis) as healthy before the event.” How can you be certain there was zero injury prior to this heatwave? Did injuries and disease not affect these octocorals prior to 2003? Please provide justification – as written now, this assumption of “pristine” octocoral populations seems unfounded.
 - Many unfamiliar terms are introduced in Material and Methods: e.g., “self-thinning”. Please introduce these concepts in the Intro so that the reader has a clear idea of why these things were measured.
 - To clarify the different models that were run and the parameters within each model, please provide model equations or a table that outlines the covariates included in each model.
- Statistical analyses:
- What type of random effects model? GLMM? GAMM? Which packages in R were used? How was model fit determined? Please provide more specifics.

- Given that there are only four different populations included in the models, and it is of interest to assess the trends within each population, it would be more appropriate to do a glm random SLOPES model (with each population's slope varying) rather than a glmm including population as a random effect.
- Seems more appropriate to assess relationship between size class and mortality by including an interaction term between size class and proportion dead/affected rather than doing all of these post-hoc tests.... Please provide justification for why the latter route was chosen.

Results

- Line 214: so there were a significant number of MHWs before 2003. Then, is it appropriate for this study to assume that that colonies were completely unaffected by heat waves prior to 2003? It seems likely that these colonies were already affected by climate change prior to this study – please acknowledge this in the paper.
- Line 238: why were injuries so high in 2016 and 2018 – were these years of extreme warming?
- Line 293: The self-thinning results seem very informative from an ecological standpoint, but I can't quite wrap my head around the logic behind the self-thinning v disturbance "line". Please provide more background and rationale for this analysis in the Material and Methods section.
- How were the trends for each site produced – with a separate model for each site? (please provide more info on model structure in Methods).

Discussion

- How is it possible to measure recovery from the 2003 marine heatwave given that subsequent heatwaves have since occurred? Could populations have been on a recovery trajectory until the next heatwave hit? Why was 2003 chosen as the baseline – was this the period after which heatwave intensity was significantly higher? Please elaborate in Intro and Discussion.
- It would be great to extrapolate the results of this study to produce some specific recommendations for candidate MPAs that account for projected climate change impacts. Are there any areas in the Mediterranean that could be considered refugia (at least for these octocoral species)? If so, where?
- Most importantly, this study does not show that MPA designation has not slowed climate change impacts, because only MPA sites were assessed. Is it possible that these octocorals had even greater heatwave-induced mortality at non-MPA sites?

Decision letter (RSPB-2020-2460.R0)

17-Nov-2020

Dear Mr Gomez Gras:

I am writing to inform you that your manuscript RSPB-2020-2460 entitled "Population collapse of habitat-forming species in the Mediterranean: a long-term study of gorgonian populations affected by recurrent marine heatwaves" has, in its current form, been rejected for publication in Proceedings B.

This action has been taken on the advice of referees, who have recommended that substantial revisions are necessary. With this in mind we would be happy to consider a resubmission, provided the comments of the referees are fully addressed. However please note that this is not a provisional acceptance.

The resubmission will be treated as a new manuscript. However, we will approach the same reviewers if they are available and it is deemed appropriate to do so by the Editor. Please note

that resubmissions must be submitted within six months of the date of this email. In exceptional circumstances, extensions may be possible if agreed with the Editorial Office. Manuscripts submitted after this date will be automatically rejected.

Sincerely,
Dr Locke Rowe
mailto:proceedingsb@royalsociety.org

Associate Editor
Board Member: 1
Comments to Author:

The two reviewers were supportive of your study and agreed that it addressed an important topic. Nevertheless, Reviewer 2 raised several major issues that will be crucial to address appropriately to reach the level of excellence needed for PRSB. I summarize them here as key requests: (1) to improve the statistical analyses (helpful suggestions are made), (2) to provide a better rationale for the study design and the analyses, and especially critical (3) to explain how you can measure recovery from the 2003 marine heat wave given that subsequent heatwaves have since occurred, and (4) to defend how your study of only Marine Protected Areas (MPAs) can conclude about MPAs not being able to slow climate change impacts. In addition, both reviewers made several other constructive suggestions to strengthen your study. Reviewer 1 asked to place your results in a broader context of studies on other Mediterranean taxa to get a better idea about generality of the observed patterns, and indicated how to improve the figures. Reviewer 2 asked to better motivate your assumption that the octocoral populations were pristine at the start of your study, and to produce some specific recommendations for candidate MPAs based on your results.

Reviewer(s)' Comments to Author:
Referee: 1

Comments to the Author(s)

The paper summarizes an interesting long-term study of the responses of marine benthic populations to recurrent heatwaves due to global warming. Although the study is spatially-limited (one marine protected area), it provides a rare insight into the long-term dynamics of structurally-important Mediterranean species. My comments are quite minor, to help improve the paper.

The paper is over-referenced, including quite a lot of self-citations. I would like to have seen a paragraph that compares these results to other studies in the Mediterranean, on coral reefs, kelp beds, etc. How general are these findings – are the same trends likely to be unfolding elsewhere?

Line 122. What does mucilaginous outbreaks mean? Disease?

Line 131. Hydrological period is unclear.

The site descriptions on Lines 136-139 are unclear. The site names appear in Fig. 2, but not in the methods. How far apart are they? Is Paluzzu cave the same as Paluzza? Were each of the four sites for *P. clavata* sampled at the same 18-26m depth range, or were some sites shallower than others? It appears to be the latter, from L39 (2.5m range). The authors sometimes seem to assume that readers will flip back and forth to the supplementary materials – but it would be better if the main text is self-explanatory.

L190. Saturated populations is unclear.

L272. Indistinctly?

The figures need improvement. Figure 1 should be taller. In Figure 2-4, the place names and cartoons on top are unnecessary. The font sizes are too small. Figure 3 should be presented as number of colonies per size class, not the proportion. The stability in proportions is misleading (L387-389), and understates the changes that have occurred. In particular, the absolute number of large, reproductive colonies must have declined substantially, compromising population resilience.

I didn't find Figure 5 very useful, and the description of it on L313-314 is inadequate. I suggest deleting this figure.

Minor points:

Some of the English could be improved. For example:

L41. ...resilience to heatwaves

L89. Parsimonious is the wrong word.

L93.which can be protracted.

L104.by monitoring 5....

L108.resilience of coralligenous....

L114.both species also exhibit late....

L220. % (here and elsewhere) should be percentage, or proportion.

L234. which coincided with a peak...

L250.neither the density nor...

L301, 303. Global?

L302.which was mostly determined by a substantial decline in density....

L347.is constrained.

L358.colonies having died....

L370. The last 4 words in the heading are not necessary.

L377.repeatedly impacted....

L387.was far from sufficient.....

Referee: 2

Comments to the Author(s)

Abstract

- Please define “coralligenous assemblages” for readers that have not encountered this term. Is this habitat limited to the Mediterranean?

Intro

- No mention of the mechanisms by which marine heat waves cause mortality of octocorals (and other habitat-forming species). Please dedicate a paragraph to outlining the physiological mechanisms of mortality.

- Line 88: “Habitat-forming species are long-lived”. But octocorals are colonial. Individual polyps are not long-lived, but the entire colony may be. Similarly, line 89 says recovery from mortality

may be slow. But what about partial mortality (i.e., when part of colony survives?). In scleractinian corals, recovery can be very fast when mortality is only partial.

- Line 92: Why does recruitment not also contribute to population recovery? Unclear – please elaborate.
- Paragraph starting at line 112: Ok, this explains that recruitment potential is low for these species – please move this information to the Intro to provide context for the comment above.
- Paragraph starting at line 120: Is this region exposed to any sources of land-based runoff or other types of pollution? Seems unlikely that any coastal area in the Mediterranean has effectively banned “any local human stressors”.
- Line 122: “mucilaginous outbreaks”? please explain.
- Please provide a couple of sentences at the end of the intro summarizing what you measured and why.

Material and Methods

- Why were the various sites chosen? Are they characterized by same environmental conditions, human population densities, etc.? What is the explanation for the unbalanced sampling design (i.e., four sites for *Paramuricea clavata* but only one site for *Corallium rubrum*?). Why were multiple sites chosen?
- Why were all these various aspects of the colonies measured (e.g., what are ecological or conservation implications of changes in density v biomass?) Please provide justification for study design before all of these different metrics are mentioned.
- Line 157: “Finally, the status of the populations in summer 2003 (before the MHW) were derived from the data obtained in May 2004, considering all tissue showing signs of recent injuries (e.g., denuded axis) as healthy before the event.” How can you be certain there was zero injury prior to this heatwave? Did injuries and disease not affect these octocorals prior to 2003? Please provide justification – as written now, this assumption of “pristine” octocoral populations seems unfounded.
- Many unfamiliar terms are introduced in Material and Methods: e.g., “self-thinning”. Please introduce these concepts in the Intro so that the reader has a clear idea of why these things were measured.
- To clarify the different models that were run and the parameters within each model, please provide model equations or a table that outlines the covariates included in each model.

Statistical analyses:

- What type of random effects model? GLMM? GAMM? Which packages in R were used? How was model fit determined? Please provide more specifics.
- Given that there are only four different populations included in the models, and it is of interest to assess the trends within each population, it would be more appropriate to do a glm random SLOPES model (with each population’s slope varying) rather than a glmm including population as a random effect.
- Seems more appropriate to assess relationship between size class and mortality by including an interaction term between size class and proportion dead/affected rather than doing all of these post-hoc tests.... Please provide justification for why the latter route was chosen.

Results

- Line 214: so there were a significant number of MHWs before 2003. Then, is it appropriate for this study to assume that that colonies were completely unaffected by heat waves prior to 2003? It seems likely that these colonies were already affected by climate change prior to this study – please acknowledge this in the paper.
- Line 238: why were injuries so high in 2016 and 2018 – were these years of extreme warming?
- Line 293: The self-thinning results seem very informative from an ecological standpoint, but I can’t quite wrap my head around the logic behind the self-thinning v disturbance “line”. Please provide more background and rationale for this analysis in the Material and Methods section.
- How were the trends for each site produced – with a separate model for each site? (please provide more info on model structure in Methods).

Discussion

- How is it possible to measure recovery from the 2003 marine heatwave given that subsequent heatwaves have since occurred? Could populations have been on a recovery trajectory until the next heatwave hit? Why was 2003 chosen as the baseline – was this the period after which heatwave intensity was significantly higher? Please elaborate in Intro and Discussion.
- It would be great to extrapolate the results of this study to produce some specific recommendations for candidate MPAs that account for projected climate change impacts. Are there any areas in the Mediterranean that could be considered refugia (at least for these octocoral species)? If so, where?
- Most importantly, this study does not show that MPA designation has not slowed climate change impacts, because only MPA sites were assessed. Is it possible that these octocorals had even greater heatwave-induced mortality at non-MPA sites?

Author's Response to Decision Letter for (RSPB-2020-2460.R0)

See Appendix A.

RSPB-2021-0780.R0

Review form: Reviewer 3

Recommendation

Major revision is needed (please make suggestions in comments)

Scientific importance: Is the manuscript an original and important contribution to its field?

Good

General interest: Is the paper of sufficient general interest?

Excellent

Quality of the paper: Is the overall quality of the paper suitable?

Acceptable

Is the length of the paper justified?

Yes

Should the paper be seen by a specialist statistical reviewer?

No

Do you have any concerns about statistical analyses in this paper? If so, please specify them explicitly in your report.

No

It is a condition of publication that authors make their supporting data, code and materials available - either as supplementary material or hosted in an external repository. Please rate, if applicable, the supporting data on the following criteria.

Is it accessible?

Yes

Is it clear?

Yes

Is it adequate?

Yes

Do you have any ethical concerns with this paper?

No

Comments to the Author

The manuscript addresses a very timely and important topic related to the occurrence and effects of marine heatwaves (MHWs) in the Mediterranean Sea. More specifically, the authors inspected the recovery (or lack thereof) of octocoral communities in a relatively long-living and well-enforced marine protected area in the north-western Mediterranean Sea. Their main result is the lack of recovery of these communities over the almost 20 years after a major MHW occurred in 2003.

I think that the topic and the results are important but I have two major concerns with the study as it has been conducted so far.

First, the authors assume that the inspected communities were in a healthy condition before the occurrence of the MHW in 2003. For example, in lines 185-187 they state that “the reference values of the populations in 2003 (before the MHW) were estimated by considering all tissue showing signs of recent injuries (e.g., denuded axis or recent epibiosis) in May 2004 as healthy before the event”. This is a very strong assumption, considering that there is ample evidence of a serious heatwave that affected octocorals in 1999 (check for example Cerrano et al 2000 Ecology Letters on the Ligurian Sea, Garrabou et al 2001 MEPS on Provence, Linares et al 2005 on Port Cros, Coma et al 2006 on Menorca). Given the extent and magnitude of this 1999 heatwave, it looks unlikely that no effect at all occurred in NW Corsica.

Second, the study is focused on populations that “exhibited clear signs of having been affected by the 2003-MHW” (lines 151-153). While I concur that such populations must be considered in the study, data on allegedly unaffected populations would have been precious as well for comparison. Have those unaffected populations by the 2003 MHW accrued damage after the multiple successive MHWs? If not, why? Which biotic or abiotic factors may have influenced such a pattern? Given the lack of any robust pre-MHW data, the comparison with such unaffected populations may have enriched the study. One of the consequences of this selection is that the number of populations used for the study is relatively small, being four for *Paramuricea clavata* and one for *Corallium rubrum* (although I wish to highlight that the number of inspected colonies is extremely high, being 801 and 64, respectively, only in 2004).

Further comments pertain to the use of SST (lines 140-142). Because the target colonies live in the subtidal, why using the sea surface temperature when most datasets have multiple depth layers that enable inspecting temperatures at the depth of occurrence of the colonies? Minimally, the authors should show that the two are highly correlated but should also explain the reasons of their choices, given that data are available.

The use of SST matched with the lack of pre-2003 data also poses another question: is temperature the main/only factor affecting colony decline? Remarkably, this topic is treated only in the legend of Figure 1 while it would certainly deserve a broader coverage in the ms.

Minor comments:

Lines 204-205: why “lack of difference” was considered an indicator of recovery? Wouldn't it be an indicator of lack of recovery?

Lines 244-245: the computation of affected colonies before the 2003 MHW is non-zero, but this apparently conflicts with the assumption that the inspected colonies were healthy before 2003 (lines 185-187).

Line 250: given the apparent lack of data before 2003, it is unclear how the number of dead colonies due to the event was computed.

Decision letter (RSPB-2021-0780.R0)

19-Jul-2021

I am writing to inform you that this version of your manuscript RSPB-2021-0780 entitled "Population collapse of habitat-forming species in the Mediterranean: a long-term study of gorgonian populations affected by recurrent marine heatwaves" has, in its current form, been rejected for publication in Proceedings B.

This action has been taken on the advice of referees, who have recommended that substantial revisions are necessary. With this in mind we would be happy to consider a resubmission, provided the comments of the referees are fully addressed. However please note that this is not a provisional acceptance.

Please find below the comments made by the referees, not including confidential reports to the Editor, which I hope you will find useful.

- 1) A 'response to referees' document including details of how you have responded to the comments, and the adjustments you have made.
- 2) A clean copy of the manuscript and one with 'tracked changes' indicating your 'response to referees' comments document.
- 3) Line numbers in your main document.
- 4) Please read our data sharing policies to ensure that you meet our requirements <https://royalsociety.org/journals/authors/author-guidelines/#data>.

Sincerely,
Dr Locke Rowe
mailto: proceedingsb@royalsociety.org

Associate Editor
Comments to Author:

A third expert reviewer had an independent, detailed look at your study and raised two major concerns, partly repeating one of the previous major concerns of Reviewer 2, on the lack of robust

pre-heat wave data on the condition of the octocorals and on the lack of ‘control sites’ not affected by the heat wave. I agree with both concerns of this Reviewer. Given that you addressed the first comment already in your resubmission it is unclear whether you will be able to convincingly address this concern.

Reviewer(s)¹ Comments to Author:

Referee: 3

Comments to the Author(s).

The manuscript addresses a very timely and important topic related to the occurrence and effects of marine heatwaves (MHWs) in the Mediterranean Sea. More specifically, the authors inspected the recovery (or lack thereof) of octocoral communities in a relatively long-living and well-enforced marine protected area in the north-western Mediterranean Sea. Their main result is the lack of recovery of these communities over the almost 20 years after a major MHW occurred in 2003.

I think that the topic and the results are important but I have two major concerns with the study as it has been conducted so far.

First, the authors assume that the inspected communities were in a healthy condition before the occurrence of the MHW in 2003. For example, in lines 185-187 they state that “the reference values of the populations in 2003 (before the MHW) were estimated by considering all tissue showing signs of recent injuries (e.g., denuded axis or recent epibiosis) in May 2004 as healthy before the event”. This is a very strong assumption, considering that there is ample evidence of a serious heatwave that affected octocorals in 1999 (check for example Cerrano et al 2000 Ecology Letters on the Ligurian Sea, Garrabou et al 2001 MEPS on Provence, Linares et al 2005 on Port Cros, Coma et al 2006 on Menorca). Given the extent and magnitude of this 1999 heatwave, it looks unlikely that no effect at all occurred in NW Corsica.

Second, the study is focused on populations that “exhibited clear signs of having been affected by the 2003-MHW” (lines 151-153). While I concur that such populations must be considered in the study, data on allegedly unaffected populations would have been precious as well for comparison. Have those unaffected populations by the 2003 MHW accrued damage after the multiple successive MHWs? If not, why? Which biotic or abiotic factors may have influenced such a pattern? Given the lack of any robust pre-MHW data, the comparison with such unaffected populations may have enriched the study. One of the consequences of this selection is that the number of populations used for the study is relatively small, being four for *Paramuricea clavata* and one for *Corallium rubrum* (although I wish to highlight that the number of inspected colonies is extremely high, being 801 and 64, respectively, only in 2004).

Further comments pertain to the use of SST (lines 140-142). Because the target colonies live in the subtidal, why using the sea surface temperature when most datasets have multiple depth layers that enable inspecting temperatures at the depth of occurrence of the colonies? Minimally, the authors should show that the two are highly correlated but should also explain the reasons of their choices, given that data are available.

The use of SST matched with the lack of pre-2003 data also poses another question: is temperature the main/only factor affecting colony decline? Remarkably, this topic is treated only in the legend of Figure 1 while it would certainly deserve a broader coverage in the ms.

Minor comments:

Lines 204-205: why “lack of difference” was considered an indicator of recovery? Wouldn’t it be an indicator of lack of recovery?

Lines 244-245: the computation of affected colonies before the 2003 MHW is non-zero, but this apparently conflicts with the assumption that the inspected colonies were healthy before 2003 (lines 185-187).

Line 250: given the apparent lack of data before 2003, it is unclear how the number of dead colonies due to the event was computed.

Author's Response to Decision Letter for (RSPB-2021-0780.R0)

See Appendix B.

RSPB-2021-2384.R0

Review form: Reviewer 3

Recommendation

Accept with minor revision (please list in comments)

Scientific importance: Is the manuscript an original and important contribution to its field?

Excellent

General interest: Is the paper of sufficient general interest?

Excellent

Quality of the paper: Is the overall quality of the paper suitable?

Good

Is the length of the paper justified?

Yes

Should the paper be seen by a specialist statistical reviewer?

No

Do you have any concerns about statistical analyses in this paper? If so, please specify them explicitly in your report.

No

It is a condition of publication that authors make their supporting data, code and materials available - either as supplementary material or hosted in an external repository. Please rate, if applicable, the supporting data on the following criteria.

Is it accessible?

Yes

Is it clear?

Yes

Is it adequate?

Yes

Do you have any ethical concerns with this paper?

No

Comments to the Author

I am here reviewing the revised version of the manuscript by Gómez-Gras et al titled "Population collapse of habitat-forming species in the Mediterranean: a long-term study of gorgonian populations affected by recurrent marine heatwaves". I have read in detail the whole manuscript

again and I have focused on the authors' response to the main weaknesses highlighted in my previous review.

I remind the Editor that the study addresses a very relevant topic related to the occurrence of Marine Heat Waves (MHWs) in the Mediterranean Sea and their effects on octocoral populations. Given the increasing evidence of the deleterious effects of climate warming on benthic communities in the basin and globally, I consider the study highly timely.

In my previous review, I highlighted three main points.

First, I suggested that the assumption that the inspected colonies were healthy before the 2003 MHW was very strong considering the evidence of a serious MHW in 1999 that negatively affected vast sectors of the north-western Mediterranean Sea.

The authors replied to this comment by providing extensive evidence of healthy colonies before 2003 based on: 1) field observations by largely the same team behind the current study, 2) in situ photographs from their 2002 vs 2003 surveys, 3) by citing published literature (e.g. Garrabou et al 2001 MEPS).

The main modifications to the manuscript have included extensive discussion of the pre-2003 conditions in part b) of the Materials and methods, the addition of figure S1 to the Electronic Supplementary Materials (ESM) and further clarifications to the text, among which I find now very clear those in lines 194-196.

Second, I remarked that the study did not include colonies unaffected by the 2003 MHW (e.g. to assess if and when the unaffected colonies degraded over the following years, therefore which drivers of degradation could be identified, etc.).

The authors very plainly and honestly admit that this data is not available as it was not envisioned in the initial study design and aims. Given the relevance of the study and the overall satisfaction with the other points I raised, I think that this remark should not prejudice the positive assessment of this revised manuscript version. It may be left as a recommendation to the investigators to further expand their design in current and future surveys.

Third, and last, I highlighted that temperature at depth, rather than at the surface (SST), would have better described environmental conditions of the studied colonies. Additionally, I remarked that the potential role of other stressors in causing the demise of those colonies was barely considered.

The authors still use SST for the quantification of the frequency and length of MHW in the study period, but adopted recorded temperature at 20 m depth to show the occurrence of values above the sublethal threshold for the monitored species in some of the studied years (lines 246-250 of the Results and Figure 1). Whereas I appreciate the use of in-situ recordings, the authors may consider the use of the Copernicus Mediterranean Sea Physics Reanalysis which is a high resolution dataset covering from 1987 to present, so almost the whole study period, and with numerous depth layers (https://resources.marine.copernicus.eu/product-detail/MEDSEA_MULTIYEAR_PHY_006_004/INFORMATION). The authors may first cross-check the consistency of this dataset with their own recordings and, if consistent, extend their time series of heat days at depth to 1987. By extending their temporal view on temperature back to the late 80s, they would include the main events of 1999 and 2003 and look for further evidence of the lack of and occurrence of, respectively, deleterious environmental conditions.

Given the strong enforcement and relatively remote location of Scandola, I concur with the authors that their additions to the discussion (lines 414-424) on the potential further causes of colony demise beyond temperature are sufficient.

Responses to the minor comments of my previous review are fully satisfying.

Further minor comments:

- the authors may reconsider the orientation of Figure 2 as now y axis labels are upside down.

Decision letter (RSPB-2021-2384.R0)

29-Nov-2021

Dear Mr Gomez Gras

I am pleased to inform you that your Review manuscript RSPB-2021-2384 entitled "Population collapse of habitat-forming species in the Mediterranean: a long-term study of gorgonian populations affected by recurrent marine heatwaves" has been accepted for publication in Proceedings B.

The referee has made some further suggestions, but acceptance of the MS is not contingent on acting on these. If you do wish to address these comments further, please be in touch with me. If not, please proof-read your manuscript carefully and upload your final files for publication. Because the schedule for publication is very tight, it is a condition of publication that you submit the revised version of your manuscript within 7 days. If you do not think you will be able to meet this date please let me know immediately.

To upload your manuscript, log into <http://mc.manuscriptcentral.com/prsb> and enter your Author Centre, where you will find your manuscript title listed under "Manuscripts with Decisions." Under "Actions," click on "Create a Revision." Your manuscript number has been appended to denote a revision.

You will be unable to make your revisions on the originally submitted version of the manuscript. Instead, upload a new version through your Author Centre.

- 1) A text file of the manuscript (doc, txt, rtf or tex), including the references, tables (including captions) and figure captions. Please remove any tracked changes from the text before submission. PDF files are not an accepted format for the "Main Document".
- 2) A separate electronic file of each figure (tiff, EPS or print-quality PDF preferred). The format should be produced directly from original creation package, or original software format. Please note that PowerPoint files are not accepted.
- 3) Electronic supplementary material: this should be contained in a separate file from the main text and the file name should contain the author's name and journal name, e.g. `authorname_procb_ESM_figures.pdf`

All supplementary materials accompanying an accepted article will be treated as in their final form. They will be published alongside the paper on the journal website and posted on the online figshare repository. Files on figshare will be made available approximately one week before the accompanying article so that the supplementary material can be attributed a unique DOI. Please see: <https://royalsociety.org/journals/authors/author-guidelines/>

4) Data-Sharing and data citation

It is a condition of publication that data supporting your paper are made available. Data should be made available either in the electronic supplementary material or through an appropriate repository. Details of how to access data should be included in your paper. Please see <https://royalsociety.org/journals/ethics-policies/data-sharing-mining/> for more details.

<http://datadryad.org/submit?journalID=RSPB&manu=RSPB-2021-2384> which will take you to your unique entry in the Dryad repository.

Once again, thank you for submitting your manuscript to Proceedings B and I look forward to receiving your final version. If you have any questions at all, please do not hesitate to get in touch.

Sincerely,
Dr Locke Rowe
<mailto:proceedingsb@royalsociety.org>

Reviewer(s)' Comments to Author:

Referee: 3

Comments to the Author(s).

I am here reviewing the revised version of the manuscript by Gómez-Gras et al titled "Population collapse of habitat-forming species in the Mediterranean: a long-term study of gorgonian populations affected by recurrent marine heatwaves". I have read in detail the whole manuscript again and I have focused on the authors' response to the main weaknesses highlighted in my previous review.

I remind the Editor that the study addresses a very relevant topic related to the occurrence of Marine Heat Waves (MHWs) in the Mediterranean Sea and their effects on octocoral populations. Given the increasing evidence of the deleterious effects of climate warming on benthic communities in the basin and globally, I consider the study highly timely.

In my previous review, I highlighted three main points.

First, I suggested that the assumption that the inspected colonies were healthy before the 2003 MHW was very strong considering the evidence of a serious MHW in 1999 that negatively affected vast sectors of the north-western Mediterranean Sea.

The authors replied to this comment by providing extensive evidence of healthy colonies before 2003 based on: 1) field observations by largely the same team behind the current study, 2) in situ photographs from their 2002 vs 2003 surveys, 3) by citing published literature (e.g. Garrabou et al 2001 MEPS).

The main modifications to the manuscript have included extensive discussion of the pre-2003 conditions in part b) of the Materials and methods, the addition of figure S1 to the Electronic Supplementary Materials (ESM) and further clarifications to the text, among which I find now very clear those in lines 194-196.

Second, I remarked that the study did not include colonies unaffected by the 2003 MHW (e.g. to assess if and when the unaffected colonies degraded over the following years, therefore which drivers of degradation could be identified, etc.).

The authors very plainly and honestly admit that this data is not available as it was not envisioned in the initial study design and aims. Given the relevance of the study and the overall satisfaction with the other points I raised, I think that this remark should not prejudice the positive assessment of this revised manuscript version. It may be left as a recommendation to the investigators to further expand their design in current and future surveys.

Third, and last, I highlighted that temperature at depth, rather than at the surface (SST), would have better described environmental conditions of the studied colonies. Additionally, I remarked

that the potential role of other stressors in causing the demise of those colonies was barely considered.

The authors still use SST for the quantification of the frequency and length of MHW in the study period, but adopted recorded temperature at 20 m depth to show the occurrence of values above the sublethal threshold for the monitored species in some of the studied years (lines 246-250 of the Results and Figure 1). Whereas I appreciate the use of in-situ recordings, the authors may consider the use of the Copernicus Mediterranean Sea Physics Reanalysis which is a high resolution dataset covering from 1987 to present, so almost the whole study period, and with numerous depth layers (https://resources.marine.copernicus.eu/product-detail/MEDSEA_MULTIYEAR_PHY_006_004/INFORMATION). The authors may first cross-check the consistency of this dataset with their own recordings and, if consistent, extend their time series of heat days at depth to 1987. By extending their temporal view on temperature back to the late 80s, they would include the main events of 1999 and 2003 and look for further evidence of the lack of and occurrence of, respectively, deleterious environmental conditions. Given the strong enforcement and relatively remote location of Scandola, I concur with the authors that their additions to the discussion (lines 414-424) on the potential further causes of colony demise beyond temperature are sufficient.

Responses to the minor comments of my previous review are fully satisfying.

Further minor comments:

- the authors may reconsider the orientation of Figure 2 as now y axis labels are upside down.

Decision letter (RSPB-2021-2384.R1)

30-Nov-2021

Dear Mr Gómez-Gras

I am pleased to inform you that your manuscript entitled "Population collapse of habitat-forming species in the Mediterranean: a long-term study of gorgonian populations affected by recurrent marine heatwaves" has been accepted for publication in Proceedings B.

Data Accessibility section

Open Access

Paper charges

Sincerely,

Proceedings B

Appendix A

Institut
de Ciències
del Mar

Dr.
Editor in Chief
Proceedings of the Royal Society B

2 June, 2021

Dear Editor,

On behalf of my co-authors and myself I am pleased to re-submit a new version of the manuscript (RSPB-2020-2460) “Population collapse of habitat-forming species in the Mediterranean: a long-term study of gorgonian populations affected by recurrent marine heatwaves” to be considered for publication in *Proceedings of the Royal Society B*.

We really appreciate the comments and changes proposed by the reviewers, which helped us to substantially improve the quality of the manuscript.

We have carefully addressed each of the concerns raised by the reviewers and incorporated all suggested changes, as listed below and as highlighted in the track-changes format file. In this new version of the manuscript, we have improved statistical analyses, provided a better rationale for the study design and the analyses, clarified that our study does not conclude (nor attempts to) that MPAs are not slowing down climate change impacts and clarified that our goal was not to measure recovery in ideal conditions (i.e., without subsequent MHWs), but in the more realistic current framework of recurrent MHWs. We have also placed our results in a broader context of studies, improved the figures and explained in detail why populations before the 2003-MHW could be considered as unaffected. With these changes, we expect that our manuscript will meet the quality expected from a scientific work published in *Proceedings of the Royal Society B*.

This manuscript has not been published and is not under consideration for publication elsewhere. We have no conflicts of interest to disclose. All authors have seen and agreed the new submitted version of the manuscript.

Please do not hesitate to contact me for any clarification or further required changes.

Sincerely,

Daniel Gómez Gras

PhD
Department of Marine Biology
Institute of Marine Sciences (ICM-CSIC), Barcelona

#Associate Editor comments

Comment:

The two reviewers were supportive of your study and agreed that it addressed an important topic. Nevertheless, Reviewer 2 raised several major issues that will be crucial to address appropriately to reach the level of excellence needed for PRSB. I summarize them here as key requests: (1) to improve the statistical analyses (helpful suggestions are made), (2) to provide a better rationale for the study design and the analyses, and especially critical (3) to explain how you can measure recovery from the 2003 marine heat wave given that subsequent heatwaves have since occurred, and (4) to defend how your study of only Marine Protected Areas (MPAs) can conclude about MPAs not being able to slow climate change impacts. In addition, both reviewers made several other constructive suggestions to strengthen your study. Reviewer 1 asked to place your results in a broader context of studies on other Mediterranean taxa to get a better idea about generality of the observed patterns, and indicated how to improve the figures. Reviewer 2 asked to better motivate your assumption that the octocoral populations were pristine at the start of your study, and to produce some specific recommendations for candidate MPAs based on your results.

Response to Associated Editor key requests:

(1) to improve the statistical analyses (helpful suggestions are made).

Following the reviewer's suggestion, we have improved the statistical analyses. We have now included new analyses that support our findings (see new Tables S4 and S5 from Electronic Supplementary Material; ESM) and included a detailed statistical section in the revised version of the manuscript from line 197-220.

(2) to provide a better rationale for the study design and the analyses, and especially critical

We have included some sentences in the introduction (lines 111-125) and in the materials and methods section (lines 150-155; lines 197-215; lines 223-231) that provide a better justification of the study design and analysis.

(3) to explain how you can measure recovery from the 2003 marine heat wave given that subsequent heatwaves have since occurred

We have carefully addressed this comment throughout the revised manuscript. It was not our intention to show that our results were representative of the recovery trends that populations could have followed under ideal conditions (i.e., if recurrent MHWs had not occurred). Nowadays, studying recovery over the long-term in absence of recurrent disturbances is no longer realistic in most parts of the world given the current global MHW trends (Smale et al. 2019). Consequently, our intention was to describe the potential trajectories that MHW-impacted populations of Mediterranean octocorals could follow in the face of climate change. That is, under recurrent MHWs occurring. We included some text to clarify this point throughout the revised version of the manuscript (i.e., lines 2-3 in the title; lines 40 and 48-49 in the Abstract; lines 121-123 in the Introduction; and especially lines 407-412 in the Discussion)

4) to defend how your study of only Marine Protected Areas (MPAs) can conclude about MPAs not being able to slow climate change impacts

We believe that there has been a misunderstanding regarding this point. Our study was not designed to address whether or not MPAs are able to slow climate change impacts in the Mediterranean. In fact, we only provide data from MPA since we don't have information from unprotected nearby sites. The issue of MPAs not being able to slow climate change impacts is critical for marine biodiversity conservation but it is beyond the scope of our study. Besides, addressing this question is complex and not straightforward as reflected by several studies reaching different conclusions about the role of MPAs face to climate change (e.g., Bruno et al. 2018; Montero-Serra et al. 2019). Thus, since this is a highly controversial topic and it is beyond the scope of our study, we have tried to be really careful in this respect. Specifically, we mention in the manuscript that:

- MHWs are driving the ecological collapse of habitat-forming octocorals within Scandola MPA, despite it is one of the oldest and best enforced throughout the Mediterranean (see lines 419-425). Thus, despite their great benefits (e.g., reducing other anthropogenic impacts in the area), and possibly slowing down climate change impacts, the MPA was not enough to prevent the collapse of the studied species.
- Our results fuel the ongoing debate (e.g., Bruno et al., 2018; Montero-Serra et al., 2019; Bates et al., 2019) about the role of current MPAs supporting the resilience of habitat-forming species facing climate change impacts (lines 420-422).
- We state that further research comparing long-term data from protected and non-protected areas is needed to clarify if current Mediterranean MPAs are enhancing the resilience of these species against MHWs to some extent (lines 422-424). This implies that octocorals could be suffering even more outside the MPAs.
- Finally, we advocate for the operationalization of a climate-responsive design and management of a fully protected network of MPAs as an ocean-based solution that could potentially prevent the collapse of the studied Mediterranean octocorals in some parts of the Mediterranean (lines 428-430).

We have addressed the other comments pointed by the Associated Editor alongside the reviewer's comments below.

Reviewer #1:

Comment 1 :

a) The paper is over-referenced, including quite a lot of self-citations.

Response:

- We agree that some of the references were redundant. Thus, we have removed those that were not adding essential information.

b) *I would like to have seen a paragraph that compares these results to other studies in the Mediterranean, on coral reefs, kelp beds, etc. How general are these findings – are the same trends likely to be unfolding elsewhere?*

Response:

To better reflect how general our results are with respect to trends unfolding elsewhere in the same and other habitat-forming species, we have added new text in the discussion from lines 352-355 and lines 413-418.

Comment 2 :

Line 122. What does mucilaginous outbreaks mean? Disease?

Response: Benthic mucilaginous outbreaks are biogenic, mass accumulations of mucous macro-aggregates that can sometimes suffocate the underlying benthic fauna. Nonetheless, since our main goal in the sentence was to transmit that we had not observed any major source of disturbance in the area apart from marine heatwaves during the monitored period, and given that mucilaginous outbreaks are just a possible example of disturbance that may not be known for most general readers, we have decided to remove “mucilaginous outbreaks” from the sentence, which now states as follows:

lines 133-135: “in the absence also of any other apparent disturbance during the study period, MHWs were detected as the most significant source of disturbance for the five monitored populations”

Comment 3:

Line 131. Hydrological period is unclear.

Response:

Since the term hydrological is unnecessary here and brings confusion, we have removed it from the sentence, which now remains as follows:

Line 143-144: “during the warm period (June to the end of November; JJASON) ...”

Comment 4:

*The site descriptions on Lines 136-139 are unclear. The site names appear in Fig. 2, but not in the methods. How far apart are they? Is Paluzzu cave the same as Paluzza? Were each of the four sites for *P. clavata* sampled at the same 18-26m depth range, or were some sites shallower than others? It appears to be the latter, from L39 (2.5m range). The authors sometimes seem to assume that readers will flip back and forth to the supplementary materials – but it would be better if the main text is self-explanatory.*

Response:

We agree that the main text should be self-explanatory. We have addressed this comment by including a new figure (New Figure 1a,b) in which the location, species, and depths of the monitored populations are clearly specified. With this information, the reader can

see that some sites were shallower than others within an overall depth range of 18-24.5 m.

Moreover, we have noticed that in the previous version of the manuscript, we had written that the deeper colonies in Palazzu were at 26m (e.g., Table S1), and thus the depth range of the studied colonies was from 18 to 26 m. However, the deeper colonies monitored in Palazzu for this study are in fact at 24.5 m. Therefore, the depth range of the monitored colonies extends from 18 to 24.5 m. We have corrected this information in the new version of the manuscript when applicable. For instance, in the new *Figure 1b* (substituting the old Table S1 and Figure S1), or in *lines 154 or 159* within the text.

Comment 5:

L190. Saturated populations is unclear.

Response:

We agree that the word “saturated” may be unclear. We have reformulated the sentence to reflect that we were referring to undisturbed populations maturing at crowded density (*line 230*). We have also substituted the word “saturated” for “undisturbed” in the panels **b** and **c** from the *new Figure 4* (previous Figure 4).

Comment 6:

L272. Indistinctly?

Response:

Since the meaning of indistinctly was not clear, we have reformulated the sentence (*lines 296-297*)

Lines 296-297: “Accordingly, there were not significant differences in the percentage of dead colonies among size classes during the 2003-MHW”

Comment 7:

The figures need improvement.

a) Figure 1 should be taller.

b) In Figure 2-4, the place names and cartoons on top are unnecessary. The font sizes are too small.

c) Figure 3 should be presented as number of colonies per size class, not the proportion. The stability in proportions is misleading (L387-389), and understates the changes that have occurred. In particular, the absolute number of large, reproductive colonies must have declined substantially, compromising population resilience.

Response:

We thank Reviewer 1 for his/her valuable suggestions. Following them, we have:

a) Increased the height of the Figure 1 (now as *Figure 1c* and with the x-y axis flipped to better show the trends while fitting in the new Figure 1 which also contains a map).

b) We have removed the place names and cartoons that appeared on top in Figures 2-4, and increased font sizes.

c) We have prepared Figures 3 and S2 (now referred as *Figure 3* and *Figure S1* respectively) considering the number of colonies per square meter in the y axis, so the reader can perceive both the loss of colonies in each size class and the overall changes in size class proportions. Moreover, we have explicitly mentioned the changes that have occurred in terms of the substantial loss of large colonies (including reproductive ones) within every population in *lines 403-406*.

Comment 8: *I didn't find Figure 5 very useful, and the description of it on L313-314 is inadequate. I suggest deleting this figure.*

Response: Following the reviewer's suggestion, we have decided to remove this figure.

Comment 9:

Some of the English could be improved. For example:

a) L41. *...resilience to heatwaves.*

Response: Corrected in (now as resilience to climate change); *line 39*.

b) L89. *Parsimonious is the wrong word.*

Response: Corrected. We have substituted "parsimonious" with "slow" in *line 91*

c) L93. *....which can be protracted.*

Response: Corrected in *Line 97*.

d) L108. *.....resilience of coralligenous....*

Response: This sentence is not present anymore in the new version

e) L114. *....both species also exhibit late....*

Response: This sentence is not present anymore in the new version

f) L220. *% (here and elsewhere) should be percentage, or proportion.*

Response: Corrected throughout the manuscript.

g) L234. *..... which coincided with a peak...*

Response: Corrected in *line 257*

h) L250. *.....neither the density nor...*

Response: Corrected in *line 274*

i) L302. *.....which was mostly determined by a substantial decline in density....*

Response: Corrected in *line 322*

j) L347. *.....is constrained.*

Response: Corrected in *line 371*

k) L358.colonies having died....

Response: Corrected in line 382

l) L370. The last 4 words in the heading are not necessary.

Response: Corrected in line 394

m) L377.repeatedly impacted....

Response: This sentence is not present anymore in the current version of the manuscript.

n) L387.was far from sufficient.....

Response: Corrected in line 399

#REVIEWER 2

Abstract

Comment 1:

Please define “coralligenous assemblages” for readers that have not encountered this term. Is this habitat limited to the Mediterranean?

Response:

We have now provided a brief definition in the abstract of coralligenous assemblages, which also indicates that they are endemic to the Mediterranean Sea. (*lines 51-52*). We have also included this information in the introduction with its associated reference (*lines 76-77*)

Introduction

Comment 2:

No mention of the mechanisms by which marine heat waves cause mortality of octocorals (and other habitat-forming species). Please dedicate a paragraph to outlining the physiological mechanisms of mortality.

Response:

Agreed. We added a paragraph outlining the physiological mechanisms of mortality (*lines 82-88*).

Comment 3:

Line 88: “Habitat-forming species are long-lived”. But octocorals are colonial. Individual polyps are not long-lived, but the entire colony may be. Similarly, line 89 says recovery from mortality may be slow. But what about partial mortality (i.e., when part of colony survives?). In scleractinian corals, recovery can be very fast when mortality is only partial.

Response:

Following’s reviewer suggestion, we have included a paragraph clarifying partial mortality on Mediterranean octocorals. When colonial Mediterranean octocorals experience partial mortality, the denuded skeletons surface is typically quickly overgrown by epibionts (e.g., filamentous algae, bryozoans, polychaetes...). The colonization by epibionts results in an extra weight and greater resistance to the water flow that leads to the breakage of affected colony parts or even the entire colony by detachment. Overall, colonies affected by partial mortality suffer a significant loss of their biomass or even their complete loss over the following months or even years from a mortality event. Thus, the recovery in the short term is hindered (or null in case of death). Besides, since the studied species have slow growth rates, their recovery from partial injuries is also protracted. We have added all this information and associated relevant references to the introduction in *lines 97-104*.

Comment 4:

Line 92: Why does recruitment not also contribute to population recovery? Unclear – please elaborate

Response:

To address this comment, we have specified that this may occur when adult mortality is high (including reproductive colonies) *lines 93-95*. We have also moved the information starting at line 112 (the information on the studied species) to the introduction, in which we refer to the data available indicating that recruitment rates are low in the studied species; *lines 90-93*.

Comment 5:

Paragraph starting at line 112: Ok, this explains that recruitment potential is low for these species – please move this information to the Intro to provide context for the comment above.

Response:

Following the reviewer's suggestion, we have moved the information starting at line 112 (the information on the studied species) to the introduction; *lines 90-93*.

Material and methods**Comment 6:**

Paragraph starting at line 120: Is this region exposed to any sources of land-based runoff or other types of pollution? Seems unlikely that any coastal area in the Mediterranean has effectively banned “any local human stressors”

Response:

We agree that it may seem unlikely that any coastal area in the Mediterranean has effectively banned any local human stressors. However, Scandola MPA is different from the vast majority of Mediterranean MPAs. Scandola is located in Corsica, an island with low industry and population density. Moreover, Scandola is located in the NW coast of the island, far away from any residual industry or large populated locations and the village closest to the Park has about 400 inhabitants. It includes a no-take zone or integral reserve where most human activities (including fishing, diving, and boat anchoring) are prohibited and where only boating and scientific diving is allowed. Consequently, pollution in Scandola coming from the coast is negligible. Scandola is, nonetheless, exposed to the diffuse pollution by plastics or nutrients that extend across the oceans.

Since we agree with the reviewer that according to the information included in the previous version of the manuscript, the banning of total human stressors may seem unlikely for readers, we have modified the paragraph to include relevant information:

1. That Scandola is located in the NW coast of Corsica, far away from any industry or dense human population (*lines 131-132*)

2. That most human activities are prohibited, with the exception of scientific diving and boating (*lines 132-133*).

Comment 7:

Line 122: “mucilaginous outbreaks”? please explain.

Response:

This point was also raised by reviewer #1, and we clarify it in the response to Comment 2, pasted below:

Benthic mucilaginous outbreaks are biogenic, mass accumulations of mucous macro-aggregates that can sometimes suffocate the underlying benthic fauna. Nonetheless, since our main goal in the sentence was to transmit that we had not observed any major source of disturbance in the area apart from marine heatwaves during the monitored period, and given that mucilaginous outbreaks are just a possible example of disturbance that may not be known for most general readers, we have decided to remove “mucilaginous outbreaks” from the sentence, which now it is as follows:

lines 133-136: “in the absence also of any other apparent disturbance during the study period, MHWs were detected as the most significant source of disturbance for the five monitored populations”

Comment 8:

Please provide a couple of sentences at the end of the intro summarizing what you measured and why.

Response:

Done. We have provided this information from *line 111-123*

Comment 9:

*Why were the various sites chosen? Are they characterized by same environmental conditions, human population densities, etc.? What is the explanation for the unbalanced sampling design (i.e., four sites for *Paramuricea clavata* but only one site for *Corallium rubrum*?). Why were multiple sites chosen?*

Response:

We have better specified the study sites, environmental conditions, and human population densities throughout the manuscript as follows:

1. We have created a new section called: “Population selection and quantitative monitoring”, in which we included a paragraph explaining why populations were selected and consequently why there is an unbalanced design (*line 149-155*)
2. We have included a new figure (*new Figure 1b*), in which the location of the populations within the Scandola MPA is shown. With this figure, the reader can see that all populations are separated by a maximum of few kilometers within the MPA and

located at similar depths (minimum of 18 – maximum of 24.5). Thus, populations can be assumed to be characterized by mostly the same environmental conditions, human densities...etc.

Comment 10:

Why were all these various aspects of the colonies measured (e.g., what are ecological or conservation implications of changes in density v biomass?) Please provide justification for study design before all of these different metrics are mentioned.

Response:

We have included some lines (lines 111-121) at the end of the Introduction in which we explain which metrics we used and why.

Comment 11:

Line 157: “Finally, the status of the populations in summer 2003 (before the MHW) were derived from the data obtained in May 2004, considering all tissue showing signs of recent injuries (e.g., denuded axis) as healthy before the event.” How can you be certain there was zero injury prior to this heatwave? Did injuries and disease not affect these octocorals prior to 2003? Please provide justification – as written now, this assumption of “pristine” octocoral populations seems unfounded.

Response:

Agreed. We have included new information to address this point. In particular, we point out that:

1. The 2003-MHW was the first severe MHW occurring in Scandola and which had evident impacts on Scandola's local biota. (line 188)
2. Signs of mortality were observed in the fall of 2003 and mortality impact surveys were conducted in the area (Garrabou et al. 2009). These surveys provided mortality data for more than 20 species, including *P. clavata* and *C. rubrum*. Therefore, we are certain about the impact of the MHW of 2003 on the studied populations (lines 189-191).
3. Our team has been conducting annual ecological surveys in the Scandola MPA since 1993. Beyond the natural mortality rates observed in this species (<10% of injured surface, Linares et al. 2008, Garrabou et al. 2009), we had never observed signs of mortality for these species in Scandola before the 2003-MHW (lines 191-195). Specifically, undisturbed populations of the monitored species typically present total mortality rates lower than 2% per year (Coma et al. 2004, Montero-Serra et al. 2019), which is in accordance to what we had observed before the event.

Comment 12:

Many unfamiliar terms are introduced in Material and Methods: e.g., “self-thinning”. Please introduce these concepts in the Intro so that the reader has a clear idea of why these things were measured.

Response:

Done. We have included this information in the intro from lines 116-121.

Comment 13:

To clarify the different models that were run and the parameters within each model, please provide model equations or a table that outlines the covariates included in each model.

Response:

Done, we have now specified this information in the statistical section of the material and methods section; lines 199-209 and in the new Tables S4 and S5, which describe the results of the models together with the covariates included in each model.

Comment 14:

What type of random effects model? GLMM? GAMM? Which packages in R were used? How was model fit determined? Please provide more specifics

Response:

We have now included a new description of the statistical analyses with more detailed and clearer descriptions in lines 197-220.

Comment 15:

Given that there are only four different populations included in the models, and it is of interest to assess the trends within each population, it would be more appropriate to do a glm random SLOPES model (with each population’s slope varying) rather than a glmm including population as a random effect.

Response:

Following the reviewer’s suggestions, and given that the responses of *P. clavata* populations may vary with time, we have now used random slope GLMMs to explore the impact and recovery trends of *P. clavata* from the 2003-MHW. This information is now specified in lines 207-209.

Comment 16:

Seems more appropriate to assess relationship between size class and mortality by including an interaction term between size class and proportion dead/affected rather than doing all of these post-hoc tests.... Please provide justification for why the latter route was chosen.

Response:

We believe that the reviewer was referring to using an interaction between time (before-after the 2003-MHW) and size class as explanatory variables to test for potential differences in the proportion of affected/dead colonies among size classes driven by the

2003-MHW. We agree that this could be a more appropriate way of testing such differences than conducting post-hoc tests in which the random effect of population is not accounted for. Therefore, we have now performed a new set of GLMMs in which we considered time*size class as independent variables and the % of affected/dead colonies as dependent variables. This information is now specified in lines 210-213

Results

Comment 17:

Line 214: so there were a significant number of MHWs before 2003. Then, is it appropriate for this study to assume that that colonies were completely unaffected by heat waves prior to 2003? It seems likely that these colonies were already affected by climate change prior to this study – please acknowledge this in the paper.

Response:

Following the reviewer's recommendations, we have acknowledged that MHWs were detected in the waters of Scandola since 2003 (line 337). However, we believe that it is still appropriate for our study to consider that the monitored populations were unaffected before 2003 in terms of MHW-induced mortality given that:

1. The 2003-MHW was the first severe MHW occurring in Scandola and which had evident impacts on Scandola 's local biota. (line 188)
2. Signs of mortality were observed in the fall of 2003 and mortality impact surveys were conducted in the area (Garrabou et al. 2009). These surveys provided mortality data for more than 20 species, including *P. clavata* and *C. rubrum*. Therefore, we are certain about the impact of the MHW of 2003 on the studied populations (lines 189-191).
3. Our team has been conducting annual ecological surveys in the Scandola MPA since 1993. Beyond the natural mortality rates observed in this species (<10% of injured surface, Linares et al. 2008, Garrabou et al. 2009), we had never observed signs of mortality for these species in Scandola before the 2003-MHW (lines 191-195). Specifically, undisturbed populations of the monitored species typically present total mortality rates lower than 2% per year (Coma et al. 2004, Montero-Serra et al. 2019), which is in accordance to what we had observed before the event.

Comment 18:

Line 238: why were injuries so high in 2016 and 2018 – were these years of extreme warming?

Response:

Since the injuries were higher than usual in both 2016, 2018 for *C. rubrum*, it is likely that during these years, MHW conditions were more severe than in other years at the depth of the monitoring. Yet, we cannot confirm this possibility based on our data, since we; i) did not look at exposure intensity (e.g., maximum temperature reached during MHWs) which can influence the response, and ii) analyzed only SST data, which may

significantly vary with respect to the temperatures at depth. We did not analyse in-situ (with depth temperatures) because our goal in this paper was not to study in detail the complex and multifactorial relationship between MHW conditions and mortality, but describing recovery patterns after the 2003-MHW in a broad context of potentially recurrent and increasing MHWs (*lines 136-140*).

Nonetheless, bearing in mind that it was not clear in our paper that MHW days obtained from SST data do not necessarily reflect the exact thermal stress suffered by the populations at the depth of the biological surveys, we have clearly acknowledged this information in the caption of *Figure 1c* (previous Figure 1) in *lines 617-621*.

Comment 19:

Line 293: The self-thinning results seem very informative from an ecological standpoint, but I can't quite wrap my head around the logic behind the self-thinning v disturbance "line". Please provide more background and rationale for this analysis in the Material and Methods section.

Response:

We have addressed this point in the following ways:

1. We have provided more background and rationale for this analysis in the Introduction section from *line 115-121* and in the material and methods from *line 223-231*.
2. We have also changed the confusing title "self-thinning vs disturbance" of that section and of the associated *Figure 4a* to "Visualization of disturbance vs recovery in species with self-thinning growth" (*line 222*), which we believe better reflects our analysis.

Comment 20:

How were the trends for each site produced – with a separate model for each site? (please provide more info on model structure in Methods).

Response:

The separate trajectories for each monitored site are not separate models but just a mere representation of their log-log biomass against density data used to visualize the trajectories of the populations with respect to their previously calculated species-specific self-thinning lines. The self-thinning lines are indeed linear models calculated using the information of biomass and density from undisturbed populations across the Mediterranean obtained from previous studies. We have now specified this information in *lines 223-231*.

Discussion

Comment 21:

a) How is it possible to measure recovery from the 2003 marine heatwave given that subsequent heatwaves have since occurred?

Response:

Unfortunately, measuring long-term recovery (decadal) from a single MHW event is not realistic given the global MHW trends, since MHWs are occurring every few years or decades in almost every location around the world (Smale et al. 2019). Consequently, our purpose was not to explore recovery in ideal conditions in this study, but to do so in a real-life framework of recurrent MHWs. Several studies (e.g., Garrabou & Harmelin, 2002; Linares et al., 2005; Montero-Serra et al. 2019...) had suggested in the past that the two studied species would have a low recovery capacity even in the long-term, given their life history traits (e.g., slow growth, late maturity, low dispersal capacity...). In our study, we have shown that in a real-life case, populations were not able to recover in 15 years, what might be the consequence of both their low recovery capacity, and the recurrent MHWs. We have now specified this information in the manuscript as follows:

1) We have stated that for long-lived marine species such as the ones monitored in our study, recovery from climatic impacts (i.e., MHWs) may depend on the sustained absence of new disturbances (*lines 121-123*)

2) We have stated that recurrent MHWs have been observed in the surface waters of Scandola after 2003, likely hindering recovery of populations (*lines 48-49 and 337*).

3) We have acknowledged that our results are therefore not necessarily representative of the trends that populations would have followed in absence of disturbance (*line 408-409*). Yet, they provide strong evidence on the potential trajectories that MHW-impacted populations of Mediterranean octocorals could follow in the face of climate change, since a maintained absence of MHWs is no longer realistic in Scandola given the currently observed MHW trends (*lines 409-412*)

b) *Could populations have been on a recovery trajectory until the next heatwave hit?*

Response:

Some signs of partial recovery were observed in the trajectories of almost every population throughout the monitoring. However, our yearly surveys showed that recovery was slow. These observations are in agreement with previous studies (e.g., Linares et al., 2005; Cerrano et al. 2005) that explored recovery capacity of these species within 2-5 years periods after a MHW. There might be recovery in terms of density in some populations, but a total recovery (to a previous level of both biomass and density), requires more time. And more time means more probability of being hit again by another MHW. In the long-term period studied here (15 years), we should have been able to see recovery to some extent. Yet, all populations followed collapse trajectories suggesting that their recovery trends may have been hindered by recurrent MHWs.

We have better specified it throughout the manuscript. For instance, in *lines 48-49* in the *Abstract* or in *lines 407-412* in the *Discussion*.

c) Why was 2003 chosen as the baseline – was this the period after which heatwave intensity was significantly higher? Please elaborate in Intro and Discussion.

Response:

2003 was chosen as the baseline because; i) it was the year of the first heatwave-induced mass mortality ever observed in Scandola (Garrabou et al. 2009) (see comments above) and ii) it was the year in which the monitoring activities (initiated with different scientific purposes) started. We have now specified this information in the Introduction (*lines 106-108*) and in the Material and methods section (*lines 150-153*)

Comment 22:

It would be great to extrapolate the results of this study to produce some specific recommendations for candidate MPAs that account for projected climate change impacts. Are there any areas in the Mediterranean that could be considered refugia (at least for these octocoral species)? If so, where?

Response:

We agree that it would be of great value to extrapolate our results to produce specific recommendations for candidate MPAs. Unfortunately, we have not yet empirically identified areas that could be refugia. We are working hard to find where these climatic refugia could be. Based on our results, and given the benefits that MPAs have for these octocoral species by reducing other anthropogenic impacts affecting them, it is clear that MPAs placed in the correct place would be fundamental for the prevalence of these species in the Mediterranean.

Comment 23:

Most importantly, this study does not show that MPA designation has not slowed climate change impacts, because only MPA sites were assessed. Is it possible that these octocorals had even greater heatwave-induced mortality at non-MPA sites?

Response:

We agree with the reviewer that this study does not show that MPA designation has not slowed climate change impact. This was not our goal in the manuscript. We clarified this point as follows:

- We state that further research comparing long-term data from protected and non-protected areas is needed to clarify if current Mediterranean MPAs are enhancing the resilience of these species against MHWs to some extent (*lines 422-424*). This implies that octocorals could be suffering the impacts of climate change even more outside the MPAs.

References

- Bates AE, Cooke RSC, Duncan MI, Edgar GJ, Bruno JF, Benedetti-Cecchi L, et al. (2019). Climate resilience in marine protected areas and the “Protection Paradox.” *Biol. Conserv.* 236, 305–314. <https://doi.org/10.1016/j.biocon.2019.05.005>
- Bruno JF, Bates AE, Cacciapaglia C, Pike EP., et al. (2018) Climate change threatens the world’s marine protected areas. *Nat. Clim. Chang.* 8, 499–503. <https://doi.org/10.1038/s41558-018-0149-2>
- Cerrano C, Arillo A, Azzini F, Calcinaï B, et al. (2005) Gorgonian population recovery after a mass mortality event. *Aquat Conserv* 15:147–157.
- Coma R, Pola E, Ribes M & Zabala M (2004) Long-term assessment of the patterns of mortality of a temperate octocoral in protected and unprotected areas: a contribution to conservation and management needs. *Ecol Appl* 14:1466–1478
- Garrabou J & Harmelin JG. (2002). A 20-year study on life-history traits of a harvested long-lived temperate coral in the NW Mediterranean: insights into conservation and management needs. *J Animal Ecol*, 71, 966–978. <https://doi.org/10.1046/j.1365-2656.2002.00661.x>
- Garrabou J, Coma R, Bensoussan N, Bally M, Chevaldonne P, Cigliano M, et al. (2009). Mass mortality in Northwestern Mediterranean rocky benthic communities: effects of the 2003 heat wave. *Global Change Biol*, 15, 1090–1103. <https://doi.org/10.1111/j.1365-2486.2008.01823.x>
- Linares C, Coma R, Diaz D, Zabala M, Hereu B & Dantart Ll. (2005). Immediate and delayed effects of a mass mortality event on gorgonian population dynamics and benthic community structure in the NW Mediterranean Sea. *Mar Ecol Prog Ser.* 305: 127-137. <https://doi.org/10.3354/meps305127>
- Montero-Serra I, Garrabou J, Doak DF, Ledoux JB & Linares C (2019) Marine protected areas enhance structural complexity but do not buffer the consequences of ocean warming for an overexploited precious coral. *J Applied Ecol* 56(5):1063–1074. <https://doi.org/10.1111/1365-2664.13321>
- Smale DA, Wernberg T, Oliver ECJ. et al. (2019). Marine heatwaves threaten global biodiversity and the provision of ecosystem services. *Nat. Clim. Chang.* 9, 306–312 <https://doi.org/10.1038/s41558-019-0412-1>

Appendix B

Dr.

Editor in Chief

Proceedings of the Royal Society B

29 October, 2021

Dear Editor,

We are pleased to re-submit the revised version of our manuscript (RSPB-2020-2460) entitled “Population collapse of habitat-forming species in the Mediterranean: a long-term study of gorgonian populations affected by recurrent marine heatwaves”.

We really appreciate the comments and changes proposed by reviewer 3, which helped us to improve the quality of the manuscript. We have carefully addressed each of the concerns raised by the reviewer and incorporated all possible suggested changes, as listed below and as highlighted in the track-changes format file. In this revised version of the manuscript, we have provided further proofs on the healthy *status* of the populations before the 2003-MHW, while also explaining more in detail how we could compute robust pre-marine heatwave data on the condition of the octocorals based on the surveys conducted immediately after the event. We have also complemented our temperature analyses with *in-situ* subtidal temperature data, updated Figure 1c accordingly, and discussed why the recurrence of warming impacts is the most likely factor leading to the observed collapse trajectories.

With these changes, we expect that our manuscript will meet the quality expected from a scientific work published in Proceedings of the Royal Society B.

This manuscript has not been published and is not under consideration for publication elsewhere. We have no conflicts of interest to disclose. All authors have seen and agreed the new submitted version of the manuscript.

Please do not hesitate to contact me for any clarification or further required changes.

Sincerely,

Dr. Daniel Gómez Gras

PhD

Department of Marine Biology

Institute of Marine Sciences (ICM-CSIC), Barcelona

Reviewer #3

Major comments:

Comment 1: *The authors assume that the inspected communities were in a healthy condition before the occurrence of the MHW in 2003. For example, in lines 185-187 they state that “the reference values of the populations in 2003 (before the MHW) were estimated by considering all tissue showing signs of recent injuries (e.g., denuded axis or recent epibiosis) in May 2004 as healthy before the event”. This is a very strong assumption, considering that there is ample evidence of a serious heatwave that affected octocorals in 1999 (check for example Cerrano et al 2000 Ecology Letters on the Ligurian Sea, Garrabou et al 2001 MEPS on Provence, Linares et al 2005 on Port Cros, Coma et al 2006 on Menorca). Given the extent and magnitude of this 1999 heatwave, it looks unlikely that no effect at all occurred in NW Corsica.*

Response:

We agree that “*the reference values of the populations in 2003 (before the MHW) were estimated by considering all tissue showing signs of recent injuries (e.g., denuded axis or recent epibiosis) in May 2004 as healthy before the event*” was a strong assumption that required further proof and/or clarification in the previous version of the manuscript. Indeed, as the reviewer noticed, this aspect is key to determine if populations were healthy before the 2003-MHW or not.

To reassure the reviewer, the research team has been conducting at least annual field trips to Scandola since 1993 and we had not observed mass mortality signs before 2003. Besides, the same research team conducted the mortality surveys during the 1999 mass mortality event (most of them authors of the mentioned references above). Thus, we can confirm that there was no effect of 1999 mass mortality event in Scandola. To clarify this point in the manuscript, we have provided further proofs and rationale to make absolutely clear in the new version that we can be sure that the recent injuries observed in May 2004 were the consequence of the 2003-MHW and that therefore populations were overall healthy before the event (< 10% average tissue necrosis per colony and < 10% of affected colonies).

Specifically:

1) We now state in *lines 199 to 201* that: “*the annual field trips carried out by members of our team in Scandola since 1993 (including 2002) revealed no sign of recent or past mass mortality before 2003*”. To provide further proof, we have now included a new supplementary figure (New Fig. S1;

see below) with the photograph (a) taken by members of our team in Scandola in 2002 (before the event). This photo clearly show how octocoral populations were in an overall healthy condition at that time, lacking any sign of recent or past mass mortality (< 10% of colonies affected and < 10% of average injured tissue per colony).

New Figure S1: Timeline of events and *status* of octocoral populations before and after the 2003-MHW. (a) General aspect of octocoral populations (i.e., *Paramuricea clavata*) in Palazzu site in autumn 2002 (before the MHW), showing healthy colonies (<10% injured colonies). (b) Same populations showing large percentages of injured colonies (i.e., recent epibiosis or denuded skeletons) after being affected by the 2003-MHW (autumn 2003). The appearance of these mortality signs following the 2003 MHW was reported by the mortality surveys conducted in the area by [26].

(c) Starting date of the long-term, quantitative monitoring conducted on the impacted populations. All photographs were taken by members of the research team involved in the field trips.

2) We have also included in the text a reference to a study that specifically assessed the impacts of the 1999-MHW in Scandola, finding no impact at all. *Lines 201 to 203: “This is in agreement with the lack of impact reported by [9] during the mortality surveys conducted to assess the impact of the 1999-MHW in the area”.*

3) It should also be noticed that in our analysis, the old injuries (i.e., old epibiosis) that colonies presented in May 2004 were not considered as healthy before the event, but as injuries that were already present before the 2003-MHW. This information was implied in the previous version of the manuscript but we have now directly stated it in the new version in *lines 194 to 198: “the reference values of the populations in 2003 (before the MHW) were estimated from the quantitative surveys of May 2004 by considering all tissue showing signs of old injuries (tissue overgrowth by organisms) as already injured before the event, and all tissue showing signs of recent injuries (e.g., denuded axis or recent epibiosis) as healthy before the event.”*

This is important because it means that all old injuries that colonies had from previous years (before 2003) were computed as injured tissue in the reference status of the populations in 2003 (before the MHW). Therefore, if populations had been severely impacted by mass mortality events in the past (e.g., the one of 1999), we would have seen a great amount of colonies affected by old injuries in Figure 2a-e. What we see, instead, is that all populations presented low levels of impact (always < 10% of affected colonies), which further suggests that populations were in an overall healthy *status* before the 2003-MHW.

4) We also state in *lines 203 to 205* that: *“the first signs of mass mortality affecting octocorals in Scandola were described to have appeared in early autumn of 2003 following the 2003-MHW [26]”.* We provide further proof by including in the new supplementary figure (*New Fig. S1*; shown above in response point 1), one more photograph (b) taken in Autumn 2003 (after the MHW) by members of our group. In this photograph, the recent impacts of the mortality event are clearly identifiable.

5) We also state in *lines 206 to 207* that; “*These signs of mass mortality were still visible in May 2004 as recent injuries (e.g., denuded axis or recent epibiosis)*”, which can be confirmed by the vast amount of recent injuries quantified in the surveys of May 2004 (> 40% of colonies with recent injuries in every population), which perfectly match with the average values reported for the region of Scandola in the rapid mortality surveys conducted immediately after the event (autumn 2003) by Garrabou et al. (2009).

Comment 2: *The study is focused on populations that “exhibited clear signs of having been affected by the 2003-MHW” (lines 151-153). While I concur that such populations must be considered in the study, data on allegedly unaffected populations would have been precious as well for comparison. Have those unaffected populations by the 2003 MHW accrued damage after the multiple successive MHWs? If not, why? Which biotic or abiotic factors may have influenced such a pattern? Given the lack of any robust pre-MHW data, the comparison with such unaffected populations may have enriched the study. One of the consequences of this selection is that the number of populations used for the study is relatively small, being four for *Paramuricea clavata* and one for *Corallium rubrum* (although I wish to highlight that the number of inspected colonies is extremely high, being 801 and 64, respectively, only in 2004).*

Response:

We agree with reviewer #3 that quantitative data on unaffected populations would have been precious for comparison. Unfortunately, we only have robust long-term quantitative data from 2003 on populations that presented clear signs of being affected by the 2003-MHW, since our initial goal at that time was to provide new insights into the long-term consequences of mass mortality events on these impacted populations. Nevertheless, we believe that our data on 5 populations still provide unique insights into the potential long-term trends that MHW-impacted octocoral populations could face in the face of climate change. Indeed, 5 out of the 5 monitored populations collapsed over the years.

Comment 3:

3a) Further comments pertain to the use of SST (lines 140-142). Because the target colonies live in the subtidal, why using the sea surface temperature when most datasets have multiple depth layers that enable inspecting temperatures at the depth of occurrence of the colonies? Minimally, the authors should show that the two are highly correlated but should also explain the reasons of their choices, given that data are available.

Response: Following the suggestion of Reviewer #3, we have now included *in-situ* subtidal T data in our analyses, and modified the panel c of new Figure 1 accordingly.

New Figure 1. (a) Map showing the location of Scandola within the Mediterranean. (b) Location, species, and depth of the five monitored populations within Scandola MPA. (c) Grey bars represent the number of MHW days occurred during the warm period (JJASSON) of each year from 1982 until 2018 in the surface waters of Scandola. Purple bars represent the number of *in-situ* extreme heat days (i.e., those with T over the 90th percentile with respect to the local climatology) over 23 °C (sublethal threshold for the monitored species; [11,12]) occurred at 20 m depth. *In-situ* T data was

available from 2004 onward (period shadowed in blue). Finally, the years with *in-situ* daily mean T reaching 25 °C (lethal threshold for the studied species; [11,12]) were marked with asterisks (*).

We had only included SST data in the previous version of the manuscript because the dataset was longer covering more years and therefore more robust to explore the long-term MHWs trends in the area. However, by also including the *in-situ* subtidal T data as suggested by #reviewer3, we now provide more detailed information on how much heat stress populations have experienced since 2003. Specifically, we show in the panel c of the new version of Figure 1 the number extreme heat days (those over the percentile 90th with respect to the interannual local climatology) over 23 °C (a sublethal threshold for the monitored species; Torrents et al. 2008; Crisci et al. 2017) experienced by the studied populations since 2003. Moreover, we also show which years exhibited daily mean temperatures over 25°C (a lethal threshold for the studied octocorals; Torrents et al. 2008; Crisci et al. 2017). We chose “extreme hot days over 23°C” rather than “*in-situ* MHW days” because at these temperatures, detrimental effects on the studied populations may occur before crossing the 5-consecutive-days threshold of the MHWs definition.

With the inclusion of the *in-situ* T data, the reader can see that the studied populations were recurrently exposed most of the years (and especially in 2009, 2016, 2017 and 2018) to heat stress conditions since 2003, This likely affected the recovery trajectories of populations.

Finally, apart from updating the panel c of Figure 1, we have added to the new version of the manuscript the text describing:

- New **legend** for new panel c Figure 1 (lines 612 to 617)
- The information regarding how the *in-situ* T data was obtained and analyzed (lines 149 to 156) in **Material and Methods** section).
- The description of the results regarding the in-situ T analysis (lines 246 to 250 in **Results** section).

3b) The use of SST matched with the lack of pre-2003 data also poses another question: is temperature the main/only factor affecting colony decline? Remarkably, this topic is treated only in the legend of Figure 1 while it would certainly deserve a broader coverage in the ms.

We agree with the reviewer that the potential main factors affecting colony decline deserved a broader coverage in the manuscript. Therefore, we have included a new paragraph in the discussion in which we discuss why temperature is the most likely factor affecting colony decline.

Lines 414-424: “Overall, little doubt remains that the collapsing trajectories reported here had their origin in the strong 2003-MHW. Yet, the underlying factors and processes explaining the posterior collapse are still a matter of debate. A variety of intrinsic (genetic, physiological...) and extrinsic (e.g., food availability, pathogens...) factors have been evocated in the past as potential modulators of the short-term and long-term ecological consequences of MHWs [e.g., 8,15,25,26,42]. Yet, the fact that the populations were recurrently exposed to heat stress since 2003 in the apparent absence of other disturbances suggests that the temperature anomalies likely played the most decisive role in preventing the recovery. In line with this hypothesis, three recurrent mass mortalities (> 10% of recently affected colonies) affecting the C. rubrum population were observed since 2003, and all occurred years with the highest heat conditions observed; 2009, 2016 or 2018.

Minor comments

Minor comment 1: Lines 204-205: why “lack of difference” was considered an indicator of recovery? Wouldn’t it be an indicator of lack of recovery?

Response: Lack of difference in the sentence; “Significant differences between 2003 and 2004 were considered as indicators of 2003-MHW impact, whereas a lack of difference thereafter (from 2005 to 2018) with respect to the 2003 initial values was considered as an indicator of recovery” is considered an indicator of total recovery of populations because we are comparing the values of the already impacted populations (from 2005 to 2018) with the reference values of 2003. In this sense, a lack of difference between 2003 and 2008 (as an example), would mean that the population in 2008 has recovered because it has reached again the same values that used to have in 2003.

Possibly, the reviewer#3 thought that we were comparing the impacted values of 2004 with the values of the posterior years. If that was the case, the lack of difference between the 2004 and the posterior values would indeed reflect lack of recovery. However, as explained above and as specified in the text, we were comparing 2003 vs 2004 for the impact, and 2003 vs the rest of years for a potential recovery to the initial reference values.

Minor comment 2: *Lines 244-245: the computation of affected colonies before the 2003 MHW is non-zero, but this apparently conflicts with the assumption that the inspected colonies were healthy before 2003 (lines 185-187).*

Response: The reason why the % of affected colonies before the 2003-MHW is not zero is because when calculating the 2003 reference values from the ones observed in May 2004 we only considered as healthy tissue before the event the tissue that presented recent injuries (i.e., denuded skeletons or recent epibiosis), which was the tissue that we could be certain that was injured during the 2003-MHW. In contrast, all tissue that in May 2004 presented old injuries (i.e., old epibiosis) was considered to be already injured when the 2003-MHW happened. This is why there are some (< 10%) colonies affected in 2003 prior to the MHW, and this is also how we can know that the populations were in an overall healthy state before the event. It is important to note that from previous studies, we know that in healthy populations there are always small proportion of affected colonies (Linares et al. 2008). Considering all the above and to avoid the previous confusion, we have now clearly specified in the text from line 194 to 198 that tissue presenting old injuries in May 2004 was considered as already injured in 2003 (before the MHW).

Lines 194 to 198: “the reference values of the populations in 2003 (before the MHW) were estimated from the quantitative surveys of May 2004 by considering all tissue showing signs of old injuries (tissue overgrowth by organisms) as already injured before the event, and all tissue showing signs of recent injuries (e.g., denuded axis or recent epibiosis) as healthy before the event.”

Minor comment 3: *Line 250: given the apparent lack of data before 2003, it is unclear how the number of dead colonies due to the event was computed.*

Response:

As explained above in the response to comment #1 and as it is now revised in the text from line 194 to 198, the 2003-reference values were computed from the 2004 surveys by considering signs of old injured tissue as already injured before the event, and recently injured tissue as healthy before the event. In this sense, 2003 (before MHW) and 2004 (after MHW) values only differ based on the levels of recent injuries and therefore any significant difference in any given population parameter between these two years reflects only the impact of the 2003-MHW.

Based on this, “the number of dead colonies due to the event” was computed by counting the number of colonies that presented in May 2004 a 100% of the tissue damaged with recent injuries (i.e., denuded skeleton or recent epibiosis). Moreover, since dead coral colonies take years in detaching from the wall, we can also be sure that we are not missing dead colonies when estimating the number of dead colonies due to the event. In May 2004, they were still all the colonies (even the dead ones) there. That is also why we are certain that there is not a lack of robust data before the 2003-MHW.

Other final adjustments

- We have updated Figure 4b,c to remove the unnecessary “line” symbol that appeared before the equations.
- Since new text was added to the main text of the new version of the manuscript, some specifics on model construction from the statistical section (*lines 210-214 in the previous version of the manuscript*) have been moved to the legends of Tables S4 and S5 in the Electronic Supplementary Material to meet article length requirements of the journal. Further, some unnecessary references have been removed.

References:

- Linares C, Coma R, Garrabou, J, Díaz D, Zabala M (2008). Size distribution, density and disturbance in two Mediterranean gorgonians: *Paramuricea clavata* and *Eunicella singularis*. *J. Appl. Ecol.* 45, 688-699. <https://doi.org/10.1111/j.1365-2664.2007.01419.x>
- Torrents O, Tambuté E, Caminiti N & Garrabou, J. (2008). Upper thermal thresholds of shallow vs. deep populations of the precious Mediterranean red coral *Corallium rubrum*: assessing the potential effects of warming in the NW Mediterranean. *J. Exp. Mar. Biol. Ecol.* 357, 7-19. <https://doi.org/10.1016/j.jembe.2007.12.006>.
- Garrabou J, Coma R, Bensoussan N, Bally M, Chevaldonne P, Cigliano M, et al. (2009). Mass mortality in Northwestern Mediterranean rocky benthic communities: effects of the 2003 heat wave. *Global Change Biol.* 15, 1090–1103. <https://doi.org/10.1111/j.1365-2486.2008.01823.x>

Pg. Marítim de la Barceloneta, 37-49
08032 Barcelona
+34 932 309 500
www.icm.csic.es

- Crisci C, Ledoux JB, Mokhtar-Jamaï K, Bally M, Bensoussan N, Aurelle D, et al. (2017). Regional and local environmental conditions do not shape the response to warming of a marine habitat-forming species. *Sci. Rep.* 7, 50-69. <https://doi.org/10.1038/s41598-017-05220-4>

MINISTERIO
DE CIENCIA
E INNOVACIÓN

CSIC
CONSEJO SUPERIOR DE INVESTIGACIONES CIENTÍFICAS

EXCELENCIA
SEVERO
OCHOA